# PaDPaF: Partial Disentanglement with Partially-Federated GANs

**Abdulla Jasem Almansoori**                    *abdulla.almansoori@mbzuai.ac.ae*
*Mohamed bin Zayed University of Artificial Intelligence*
*Abu Dhabi, UAE*

**Samuel Horváth**                    *samuel.horvath@mbzuai.ac.ae*
*Mohamed bin Zayed University of Artificial Intelligence*
*Abu Dhabi, UAE*

**Martin Takáč**                    *martin.takac@mbzuai.ac.ae*
*Mohamed bin Zayed University of Artificial Intelligence*
*Abu Dhabi, UAE*

**Reviewed on OpenReview:** *https://openreview.net/forum?id=vsez76EAV8*

## Abstract

Federated learning has become a popular machine learning paradigm with many potential real-life applications, including recommendation systems, the Internet of Things (IoT), healthcare, and self-driving cars. Though most current applications focus on classification-based tasks, learning personalized generative models remains largely unexplored, and their benefits in the heterogeneous setting still need to be better understood. This work proposes a novel architecture combining global client-agnostic and local client-specific generative models. We show that using standard techniques for training federated models, our proposed model achieves privacy and personalization by implicitly disentangling the globally consistent representation (i.e. *content*) from the client-dependent variations (i.e. *style*). Using such decomposition, personalized models can generate locally unseen labels while preserving the given style of the client and can predict the labels for all clients with high accuracy by training a simple linear classifier on the global content features. Furthermore, disentanglement enables other essential applications, such as data anonymization, by sharing only the content. Extensive experimental evaluation corroborates our findings, and we also discuss a theoretical motivation for the proposed approach.

## 1 Introduction

Federated learning (FL) (Konečný et al., 2016a;b) is a recently proposed machine learning setting where multiple clients collaboratively train a model while keeping the training data decentralized, i.e. local data are never transferred. FL has potential for the future of machine learning systems as many machine learning problems cannot be efficiently solved by a single machine for various reasons, including scarcity of data, computation, and memory, as well as adaptivity across domains.

Most of the focus of federated learning has been on classification models, mainly for decision-based applications. These applications include a wide variety of problems such as next-word prediction, out-of-vocabulary word and emoji suggestion, risk detection in finance, and medical image analysis (Wang et al., 2021; Kairouz et al., 2019). However, based on recent works on representation learning and causality (Schölkopf et al., 2021; Wang & Jordan, 2021; Mouli & Ribeiro, 2022), we believe that in order to understand the causes, explain the process of making a decision, and generalize it to unseen domains of data, it is crucial, if not necessary, to

learn a generative model of the data. A representation of the data (e.g. its description or features) can allow us to make decisions directly from it without seeing the data itself. For example, a description of what a digit looks like is sufficient to infer a previously unseen label. Therefore, by collaboratively learning a generative model and communicating with other clients having the same representations, the clients can potentially generalize to the unseen domains of the other clients. In general, the potential of a group of specialized machines that can communicate efficiently could be superior to that of a single monolith machine that can handle every task. These insights motivated us to propose the ideas presented in our work.

We take a step towards this objective and introduce our core idea, which can be described as follows: *to learn generative models of heterogeneous data sources collaboratively and to learn a globally consistent representation of the data that generalizes to other domains.* The representation should contain sufficient information to classify the data content (e.g. label) with reasonable accuracy, and the generative model should be able to generate data containing this content under different domains (e.g. styles). Thus, we make no assumptions about the availability of labels per client and the intersection of the clients' data distributions.

A direct application of our model is to disentangle the latent factors that correspond to the content and learn a simple classifier on top of it. This approach tackles the problem of domain adaptation (Zhang et al., 2013), where, in our case, the domain is the client's local distribution. In this scenario, our model can learn a client-agnostic classifier based on the global representation, which should generalize across all clients. We call the client-agnostic part of the representation–*the content*, and the private part–*the style*, so we partially disentangle the latent factors into content and style factors (Kong et al., 2022). Furthermore, the generator's data samples from the client's distribution can be used for data augmentation or as synthetic data for privacy. We can also actively remove private information or any client-identifying variations from the client's data by introducing a "reference" client that trains only on standardized, publicly available data with no privacy concerns. This way, we can anonymize private data by extracting its content and generating a sample with the same content but with the reference client's style.

The generative model we use in this work is Generative Adversarial Networks (GANs) (Goodfellow et al., 2020). In particular, we use the style mapping idea from StyleGAN (Karras et al., 2019). In the case of supervised data, we condition the GAN using a projection discriminator (Miyato & Koyama, 2018) with spectral normalization (Miyato et al., 2018) and a generator with conditional batch normalization (De Vries et al., 2017). We believe that designing more sophisticated architectures can improve our results and make them applicable to a broader variety of problems.

**Contributions.** Below, we summarize our contributions:

- We propose our framework for Partial Disentanglement with Partially-Federated Learning Using GANs. Only specific parts of the model are federated so that the representations learned by the GAN's discriminator can be partially disentangled into global and local factors, where we are mainly concerned with the global factors, which can be used for inference. We further enforce this disentanglement through a "contrastive" regularization term based on the recently proposed Barlow Twins (Zbontar et al., 2021).
- Our model can learn a globally consistent representation of the content of the data (e.g. label) such that it classifies the labels with high accuracy. In the case of supervised data, our model can generalize to locally unseen labels in terms of classification and generation.
- Through extensive experimentation, we validate our claims and show how popular techniques from federated learning, GANs, and representation learning can seamlessly blend to create a robust model for both classifications and generation of data in real-world scenarios.
- We make the code publicly available for reproducibility at `https://github.com/zeligism/PaDPaF`.

**Organization.** Our manuscript is organized as follows. Section 2 talks about related work and the novelty of our approach. Section 3 describes notations and preliminary knowledge about the frameworks used in our model. Section 4 contains the main part of our paper and describes the model design and training algorithm in detail. Section 5 shows a detailed evaluation of the model's generative capabilities across different domains, robustness under limited label availability, and generalization capabilities based on its learned representation. Section 6 concludes the manuscript by providing exciting future directions and summarizing the paper.

## 2 Related Work

Training GANs in the FL setting is not new. For example, (Fan & Liu, 2020) considers the options of averaging either the generator or the discriminator, but their training algorithm has no client-specific components or disentanglement. Similarly, FedGAN (Rasouli et al., 2020) also proposes a straightforward federated algorithm for training GANs and proves its convergence for distributed non-iid data sources. Another work exploring distributed GANs with non-iid data sources is (Yonetani et al., 2019), which is interesting because the weakest discriminator can help stabilize training. Still, in general, their framework does not align with ours.

The idea of splitting the model into two or more sub-modules where one focuses on personalization or local aggregation has been explored before under the name of Split Learning. Very few of these works consider a generative framework, which is the core consideration of our work. For example, FedPer (Arivazhagan et al., 2019) and ModFL (Liang et al., 2022) suggest splitting the model depth-wise to have a shared module at the beginning, which is trained with FL, and then a personalization module is fine-tuned by the client. Inversely, (Vepakomma et al., 2018) personalize early layers to provide better local data privacy and personalization. A more recent work (Pillutla et al., 2022) proposes the same idea of partial personalization based on domain expertise (e.g. in our case, setting style-related modules for personalization). HeteroFL (Diao et al., 2020) is another framework for addressing heterogeneous clients by assigning different levels of "locality" where the next level aggregates all parameters, excluding ones from the previous levels so that higher levels are subsets contained in the earlier levels. Further, (Horváth et al., 2021) trains orderly smaller subsets of weights for devices with fewer resources. This is similar in spirit, as we aim to decompose the parameters based on how they are federated.

Splitting parameters is one way of personalizing FL methods. Another well-known approach is Ditto (Li et al., 2021a), which trains a local model with a proximal term that regularizes its distance to a global model trained with FedAvg (without the regularizer). Similarly, FedProx (Li et al., 2018) adds the proximal regularization in the global objective without maintaining a local state. Other approaches work by clustering similar clients (Ghosh et al., 2020; Werner et al., 2023) or assuming a mixture of distributions (Marfoq et al., 2021; Wu et al., 2023), which can be relevant to our framework when many clients share the same style.

Regarding the privacy of federated GANs, (Augenstein et al., 2020) considers a setup where the generator can be on the server as it only needs the gradient from the discriminator on the generated data. However, they are mainly concerned with resolving the issue of debugging ML models on devices with non-inspectable data using GANs, which is a different goal from ours. Though they mention that the generator can be deployed entirely on the server, the discriminator still needs direct access to data.

Combining GANs with contrastive learning has also been explored before. ContraGAN (Kang & Park, 2020) is a conditional GAN model that uses a conditional contrastive loss. Our model is not necessarily conditional, and we use a contrastive loss as a regularizer and train GANs regularly. Another work (Yu et al., 2021) trains GANs with attention and a contrastive objective instead of the regular GAN objective, which is again different from our implementation. Perhaps the most relevant is ContraD (Jeong & Shin, 2021), a contrastive discriminator that learns a representation from which parts are fed to different projections for calculating a GAN objective and a contrastive objective. Our work differs in many details, where the main similarity to prior work is the usage of contrastive learning on a discriminator representation.

## 3 Preliminaries

This section describes the frameworks we employ to design and train our model, namely federated learning and generative models. We will also explain the notations we use in detail to remove ambiguity. Generally, we follow prevalent machine learning notation and standard notations used in federated learning (Wang et al., 2021) and GANs (Miyato & Koyama, 2018).

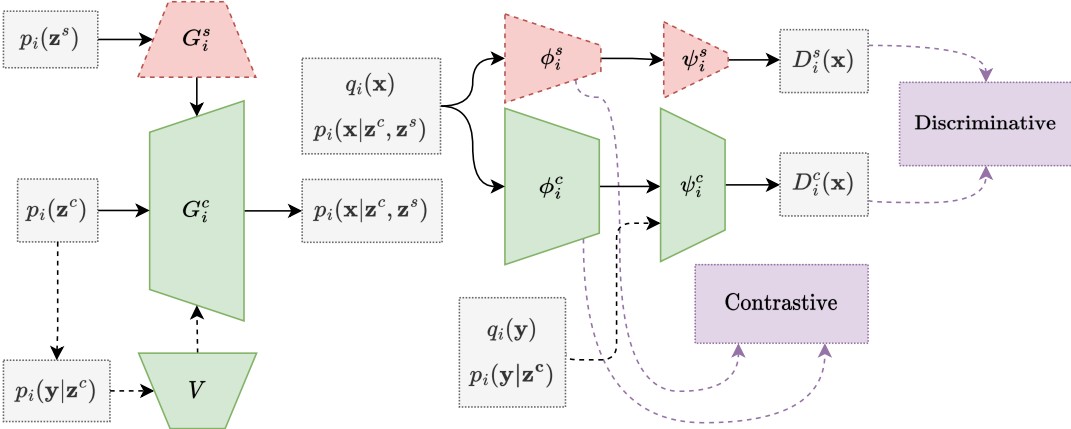

Figure 1: The architecture of our model. Federated modules are aggregated in each communication round (shown in green, solid blocks), whereas private modules are not (shown in red, dashed blocks). The *discriminative* loss is just the GAN loss as described in Sec. 4.1, whereas the *contrastive* regularization term is explained in more detail in Sec. 4.3. We describe the model's components in full detail in the Appendix.

## 3.1 Federated Learning

Federated learning is a framework for training massively distributed models, where a server typically orchestrates the training that happens locally on the models deployed on user devices (i.e. clients) (Kairouz et al., 2019). The most widely used algorithm for training models in such a setting is `FedAvg` (McMahan et al., 2016), which operates by running stochastic gradient descent on the client's models for a few steps and then aggregating the updates on the server. Another popular algorithm that accounts for data heterogeneity is `FedProx` (Li et al., 2018), which adds a proximal term to the objective that regularizes the local model to be close to the global model instead of aggregation.

The goal of federated learning is to optimize the following objective

$$F(\theta) = \mathop{\mathbb{E}}_{i \sim \mathcal{P}}[F_i(\theta)], \quad F_i(\theta) = \mathop{\mathbb{E}}_{\xi \sim \mathcal{D}_i}[f_i(\theta; \xi)], \tag{1}$$

where $\theta \in \mathbb{R}^d$ is the parameter, $\mathcal{P}$ is the client distribution, $i$ is the index of the sampled client, and $\xi$ is a random variable of the local distribution $\mathcal{D}_i$ of client $i$. Due to the nature of our framework, we also denote $\theta_i$ as the parameters at client $i$. We reserve the client index 0 for the server. For example, the client distribution $\mathcal{P}$ could be proportional to the size of the local dataset. The local distribution $\mathcal{D}_i$ could be a minibatch distribution on the local dataset $\{(\mathbf{x}_j, \mathbf{y}_j)\}_{j=1}^{N_i}$, so that $\xi \sim \mathcal{D}_i$ is a subset of indices of a minibatch.

Note that the notation for the distributions in this work will be based on the generative model literature to avoid confusion. Namely, $q_i(\mathbf{x}, \mathbf{y}) = q(\mathbf{x}, \mathbf{y}|i)$ is the target (data) distribution for client $i$ and $p_i(\mathbf{x}, \mathbf{y}; \theta)$ is the generative model for client $i$, where we often hide the dependence on $\theta$ for clarity.

**Private Modules.** Our model uses private modules or parameters, as in (Bui et al., 2019). Namely, for each client $i$, we split the parameters of its model $\theta_i$ into a *private* part $\theta_i^{\text{pvt}}$ and a *federated* part $\theta_i^{\text{fed}}$, typically in a non-trivial, structured manner that is specific to the model design. The only difference between these two parts is that the federated parameters are aggregated, whereas the private parameters are kept as is.

## 3.2 Generative Adversarial Networks

Generative Adversarial Networks (GANs) (Goodfellow et al., 2020) are powerful generative models that are well-known for their high-fidelity image generation capabilities (Karras et al., 2019) among other applications. The framework consists of two competing models playing a minimax game: a *generator* and a *discriminator*. The generator's main objective is to generate samples that cannot be discriminated from real samples.

To put it formally, let the true distribution over the data be $q(\mathbf{x})$, the generative model be $G : \mathcal{Z} \to \mathcal{X}$ and the discriminator be $D : \mathcal{X} \to \mathbb{R}$, for some data space $\mathcal{X}$ and latent space $\mathcal{Z}$. Further, let $p(\mathbf{z})$ be the latent prior

distribution, which is typically standard normal $\mathcal{N}(0, I)$. Thus, the GAN objective is $\min_G \max_D V(D, G)$ where

$$V(D, G) = \mathop{\mathbb{E}}_{q(\mathbf{x})} \log(D(\mathbf{x})) + \mathop{\mathbb{E}}_{p(\mathbf{z})} \log(1 - D(G(\mathbf{z}))). \tag{2}$$

**Conditional GANs.** If labels $\mathbf{y}$ are available, the data space becomes $\mathcal{X} \times \mathcal{Y}$, where $\mathcal{Y}$ can be $\{0, 1\}^{d_y}$ for a set of $d_y$ classes, for example. We can define a generative distribution over data $G(\mathbf{z}) = p(\mathbf{x}, \mathbf{y}|\mathbf{z})p(\mathbf{z})$, where we can factor $p(\mathbf{x}, \mathbf{y}|\mathbf{z}) = p(\mathbf{x}|\mathbf{y}, \mathbf{z})p(\mathbf{y}|\mathbf{z})$, for example. So, we can rewrite the GAN objective with an implicit generator as follows

$$V(D, G) = \mathop{\mathbb{E}}_{q(\mathbf{y})q(\mathbf{x}|\mathbf{y})} \log(D(\mathbf{x}, \mathbf{y})) + \mathop{\mathbb{E}}_{p(\mathbf{y})p(\mathbf{x}|\mathbf{y})} \log(1 - D(\mathbf{x}, \mathbf{y})). \tag{3}$$

The method we adopt for conditioning is the projection discriminator (Miyato & Koyama, 2018). First, let us examine the unconditional case in more detail. We can write $D(\mathbf{x}) = (\mathcal{A} \circ \psi \circ \phi)(\mathbf{x})$ a decomposition of the discriminator, where $\mathcal{A}$ is the activation (e.g. $\mathcal{A}$ is the sigmoid function in (3)), $\psi : \mathcal{R} \to \mathbb{R}$ is a linear layer, and $\phi(\mathbf{x}) : \mathcal{X} \to \mathcal{R}$ returns the features of $\mathbf{x}$. Note that $\mathcal{A} = \text{sigmoid}$ implies $D(\mathbf{x}) = p(\mathbf{x})$.

Then, the basic idea of adding a conditional variable $\mathbf{y}$ is to model $p(\mathbf{y}|\mathbf{x})$ as a log-linear model in $\phi(\mathbf{x})$

$$\log p(\mathbf{y} = y|\mathbf{x}) := (\boldsymbol{v}_y^p)^\mathsf{T} \phi(\mathbf{x}) - \log Z(\phi(\mathbf{x})), \tag{4}$$

where $Z(\phi(\mathbf{x})) := \sum_{y' \in \mathcal{Y}} \exp\left((\boldsymbol{v}_{y'}^p)^\mathsf{T} \phi(\mathbf{x})\right)$ is the partition function and $\boldsymbol{v}_y^p$ is an embedding of $y$ w.r.t. $p$. The authors observed that the optimal $D$ has the form $D^* = \mathcal{A}(\log q^*(\mathbf{x}, \mathbf{y})/\log p^*(\mathbf{x}, \mathbf{y}))$ (Goodfellow et al., 2020), so given (4) and an embedding $V$, they reparameterized $D$ as

$$D(\mathbf{x}, \mathbf{y}) = \mathcal{A}\left(\mathbf{y}^\mathsf{T} V \phi(\mathbf{x}) + \psi(\phi(\mathbf{x}))\right). \tag{5}$$

The choice $\mathcal{A} = \text{sigmoid}$ is used in the vanilla GAN objective, and $\mathcal{A} = \text{id}$ is used for the Wasserstein metric (Gulrajani et al., 2017). We use the latter in our implementation.

### 3.3 Self-Supervised Learning

Self-supervised learning (SSL) is a modern paradigm in which the supervision is implicit, either by learning a generative model of the data itself or by leveraging some invariance in the data (Liu et al., 2021). One well-known method uses a contrastive objective (Le-Khac et al., 2020), which pulls similar data together and pushes different data away.

Let us describe a general contrastive loss, which includes InfoNCE (Oord et al., 2018) and NT-Xent (Chen et al., 2020). Let $\phi$ be the feature of $\mathbf{x}$, and let $\mathbf{x}$ and $\mathbf{x}^+$ be two similar data. Then, given a similarity metric "sim", e.g. the inner product, and a temperature $\tau$, we have

$$\mathcal{L}_{\text{contrast}}(\mathbf{x}; \mathbf{x}^+) = -\log \frac{\exp(\text{sim}(\phi(\mathbf{x}), \phi(\mathbf{x}^+))/\tau)}{\mathbb{E}_{\mathbf{x}^-}[\exp(\text{sim}(\phi(\mathbf{x}), \phi(\mathbf{x}^-))/\tau)]}. \tag{6}$$

The need for mining $\mathbf{x}^-$ has been mitigated with recent advances such as moving-average encoders (He et al., 2020) or stop gradient operators (Chen & He, 2021). One technique of particular interest is Barlow Twins (Zbontar et al., 2021), which optimizes the empirical cross-correlation of two similar features to be as close to identity as possible[1]. The empirical cross-correlation between two batches of (normalized) projected features $\mathbf{A} = g(\phi(\mathbf{x}))$ and $\mathbf{B} = g(\phi(\mathbf{x}^+))$ for some projection $g$ is defined as

$$\mathcal{C}(\mathbf{A}, \mathbf{B})_{ij} = \frac{\sum_b \mathbf{A}_{b,i} \mathbf{B}_{b,j}}{\sqrt{\sum_b (\mathbf{A}_{b,i})^2} \sqrt{\sum_b (\mathbf{B}_{b,j})^2}} \tag{7}$$

Now, the objective of Barlow Twins is

$$\mathcal{L}_{\text{BT}}(\phi(\mathbf{x}), \phi(\mathbf{x}^+)) = \sum_i (1 - \mathcal{C}_{ii})^2 + \lambda_{BT} \sum_{i \neq j} \mathcal{C}_{ij}^2, \tag{8}$$

where we omit the dependence of $\mathcal{C}$ on $g(\phi(\mathbf{x}))$ and $g(\phi(\mathbf{x}^+))$ for clarity. The hyper-parameter $\lambda_{BT}$ controls the sensitivity of the off-diagonal "redundancy" terms and is often much smaller than 1.

---

[1]Which maximizes correlation and minimizes redundancy, thus the name "Barlow Twins" (Barlow, 2001).

# 4 Partial Disentanglement with Partially-Federated GANs

In this section, we present a framework for **Pa**rtial **D**isentanglement with **Pa**rtially-**F**ederated (PaDPaF) GANs, which aims to recover the client invariant (content) representation by disentangling it from the client-specific (style) variation. This is done by a contrastive regularization term that encourages the discriminator to learn a representation invariant to variations in some latent subspace, i.e., given a content latent, the representation should be invariant to variations in style latent.

## 4.1 Federated Conditional GANs

Starting from the GAN framework in Sec. 3.2, we aim to optimize (3) in the federated setting with respect to the parameters of $D_i$ and $G_i$ per client $i$. The discriminator architecture is based on (He et al., 2016) with the addition of spectral normalization (Miyato et al., 2018). In addition to the global discriminator, we also introduce a smaller, local version of the global discriminator for each client. The global generator is a style-based ResNet generator following Gulrajani et al. (2017) with the addition of a local style vectorizer for each client (Karras et al., 2019). The generator is conditioned on the output of the style vectorizer through conditional batch normalization layers (De Vries et al., 2017) that are kept local too (Li et al., 2021b).

This split between local and global parameters is done in a way such that the domain-specific parameters are private and solely optimized by the clients' optimizers, whereas the "content" parameters are federated and optimized collaboratively. Formally, for each client $i$, we introduce $D_i^c(\mathbf{x}, \mathbf{y}) : \mathcal{X} \times \mathcal{Y} \to \mathbb{R}$, the "content" discriminator, and $D_i^s(\mathbf{x}) : \mathcal{X} \to \mathbb{R}$ is the "style" discriminator. Similarly, let the content (global) generator be $G_i^c(\mathbf{z}^c; \mathbf{s}_i, \mathbf{y}) : \mathcal{Z}^c \times \mathcal{S}_i \times \mathcal{Y} \to \mathcal{X}$ and the the style vectorizer be $G_i^s : \mathcal{Z}_i^s \to \mathcal{S}_i$, so that $\mathbf{s}_i = G_i^s(\mathbf{z}^s)$. For simplicity, we assume that $D_i^c$ and $D_i^s$ have the same architectures, but this is not necessary. Let the content featurizer be $\phi_i^c : \mathcal{X} \to \mathbb{R}^{d_c}$, and similarly the style featurizer $\phi_i^s : \mathcal{X} \to \mathbb{R}^{d_s}$. For simplicity, we can assume that $d_s = d_c$. Note that $\phi_i^c$ is a part of $D_i^c$, and $\phi_i^s$ is a part of $D_i^s$, as explained in Sec. 3.2. We make the dependence of these modules on their parameters $\theta$ implicit unless stated otherwise. Now, we rewrite the client objective to follow the convention in (1)

$$f_i(\theta; \xi) = \mathop{\mathbb{E}}_{\mathbf{x}, \mathbf{y} \in \xi} \left[ \log(D_i^c(\mathbf{x}, \mathbf{y}) + \log(D_i^s(\mathbf{x}))\right] + \mathop{\mathbb{E}}_{\tilde{\mathbf{x}}, \tilde{\mathbf{y}} \sim p_i(\cdot; \theta)} \left[\log(1 - D_i^c(\tilde{\mathbf{x}}, \tilde{\mathbf{y}})) + \log(1 - D_i^s(\tilde{\mathbf{x}}))\right], \qquad (9)$$

where $\xi \sim q_i(\mathbf{x}, \mathbf{y})$. Here, we make the dependence on $\theta$ explicit for the generative model $p_i$ because $p_i(\tilde{\mathbf{x}}, \tilde{\mathbf{y}}; \theta) = G_i^c(\mathbf{z}^c, G_i^s(\mathbf{z}_i^s), \tilde{\mathbf{y}})$ where $\mathbf{z}^c, \mathbf{z}_i^s \sim \mathcal{N}(0, I)$ and $\tilde{\mathbf{y}} \sim \mathrm{Unif}(\mathcal{Y})$.

## 4.2 Partially-Federated Conditional GANs

When the client's data distributions are not iid—which is the case in practice—the client's domain $i$ can act as a confounding variable. In our case, we assume that each client has its own domain and unique variations, as in a cross-silo setting. In practice, especially when we have millions of clients, we can cluster the clients into similar domains and treat them as silos. The implementation of this idea will be left for future work.

Before we proceed, we make a simplifying assumption about the content of $\mathbf{y}$.

**Assumption 1 (Independence of content and style)** *Given $\mathbf{x} \sim q(\mathbf{x})$, the label $\mathbf{y}$ is independent from client $i$, i.e. we gain no information about $\mathbf{y}|\mathbf{x}$ from knowing $i$. Namely,*

$$q_i(\mathbf{y}|\mathbf{x}) = q(\mathbf{y}|\mathbf{x}). \qquad (10)$$

It is worth mentioning that this assumption might not be entirely true in some practical situations, particularly when "personalization" is needed. Indeed, label prediction personalization helps exactly when our assumption does not hold. For example, in the case of MNIST, where client A writes the digits 1 and 7 very similarly to each other, and client B writes 1 similarly to client A but writes 7 with a crossing slash such that it is perfectly distinguishable from 1. Then, a sample of the digit 7 without a slash would be hard to distinguish from 1 for client A but easily recognizable as 1 for client B. Hence, given $\mathbf{x}$, there could be some information in $i$ that can help us predict $\mathbf{y}$, which is what personalization makes use of. However, we make the simplifying

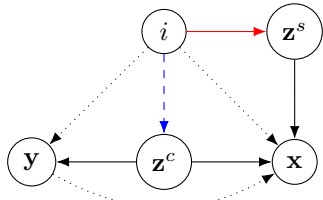

Figure 2: A causal model with the generative latent variables $\mathbf{z}^c$ and $\mathbf{z}^s$ of $\mathbf{x}$. The dotted arrows show the underlying causal model without latents. Both latents depend on the client $i$, and we assume that the content latent $\mathbf{z}^c$ generates the label $\mathbf{y}$ as well. The blue arrows drop when we $do(\mathbf{z}^c)$, and the red arrows drop when we $do(\mathbf{z}^s)$. The dashed arrows illustrate the reduced influence of $i$ over the mechanism $i \to \mathbf{z}^c$ as we run federated averaging. $\mathbf{z}^s$ and its mechanism $i \to \mathbf{z}^c$, on the other hand, remain specific to the client in our case.

assumption that the global representation of the content of $\mathbf{x}$ should be, in theory, independent from the local variations introduced by the client.

Given this assumption, we would like to construct our base generative model such that $p_i(\mathbf{y}|\mathbf{x}) := p(\mathbf{y}|\mathbf{x}, i)$ is invariant to $i$ and $p_i(\mathbf{x}) := p(\mathbf{x}|i)$ generates data from domain $i$. Let $p(i)$ be the probability of sampling client $i$, which we assume to be proportional to the number of data in client $i$, and $p(\mathbf{y})$ the prior of the labels, which we also assume to be proportional to the global occurrences of $\mathbf{y}$. Note here that the label distribution for a given $i$ could be disjoint or has a small overlap with different $i$'s. This is particularly true in practice, as clients might not have access to all possible labels. More importantly, it implies that $\mathbf{x}$ can depend non-trivially on $i$ as well, e.g. through data augmentations (more details on the construction in the Appendix). However, we want our model to be agnostic to the client's influence on $\mathbf{y}|\mathbf{x}$ for a globally consistent inference while being specific to its influence on $\mathbf{x}$ for data generation personalization.

We enforce the independence of $\mathbf{y}|\mathbf{x}$ from $i$ in $p_i(\mathbf{y}|\mathbf{x})$ via our model construction and split of private and federated parameters. This split influences the learned generative mechanism $\mathbf{z}^c, \mathbf{z}^s \to \mathbf{x}$. The generative mechanism from $\mathbf{z}^s \to \mathbf{s}$ is private and dependent on $i$, while the generative mechanism from $\mathbf{z}^c, \mathbf{s} \to \mathbf{x}$ is also dependent on $i$ but becomes independent as we run federated averaging. The causal model we assume is shown in Fig. 2.

**Generative vs. discriminative.** We can write the generative probability as

$$q(\mathbf{y}|\mathbf{z}^c)q(\mathbf{x}|\mathbf{z}^c, \mathbf{z}^s)q(\mathbf{z}^c|i)q(\mathbf{z}^s|i)q(i), \tag{11}$$

which describes the generation process starting from the right-most term. We can also factor the probability distribution in a discriminative form as

$$q(\mathbf{z}^c|\mathbf{x}, \mathbf{y}, i)q(\mathbf{z}^s|\mathbf{x}, i)q(\mathbf{x}, \mathbf{y}|i)q(i), \tag{12}$$

which also describes the discriminative process from right to left. Note that we have access to the sampling distribution $q(i)$, local datasets $q(\mathbf{x}, \mathbf{y}|i)$, and latent distributions $q(\mathbf{z}^c|i)$ and $q(\mathbf{z}^s|i)$. It remains to estimate the generators $q(\mathbf{y}|\mathbf{z}^c)$ and $q(\mathbf{x}|\mathbf{z}^c, \mathbf{z}^s)$ and the encoders/discriminators $q(\mathbf{z}^c|\mathbf{x}, \mathbf{y}, i)$ and $q(\mathbf{z}^s|\mathbf{x}, i)$ with the model $p(\cdot; \theta)$. We can generate $p(\mathbf{x}|\mathbf{z}^c, \mathbf{z}^s)$ in a straightforward manner with GAN training and federated averaging. We can also generate $p(\mathbf{y}|\mathbf{z}^c)$ by sampling labels uniformly. This is possible because we can augment $\mathbf{z}^c$ with a variable $\tilde{\mathbf{y}} \sim \text{Unif}(\mathcal{Y})$, and then set $\mathbf{y} := \tilde{\mathbf{y}}$. Recall (4) and note that we maximize $p(\mathbf{y}|\mathbf{x})$ as a log-linear model in the global features $\phi^c(\mathbf{x})$. We now make another simplifying assumption.

**Assumption 2 (Feature to latent map)** *There exist functions $\mathfrak{Z}^c : \mathbb{R}^{d_c} \times \mathcal{Y} \to \mathcal{Z}^c$ and $\mathfrak{Z}^s : \mathbb{R}^{d_s} \to \mathcal{Z}^s$ such that*

$$\mathfrak{Z}^c(\phi^c(\mathbf{x}), \mathbf{y}) \approx \mathbf{z}^c, \quad \text{and} \quad \mathfrak{Z}^s(\phi_i^s(\mathbf{x})) \approx \mathbf{z}^s. \tag{13}$$

The functions $\mathfrak{Z}^c$ and $\mathfrak{Z}^s$ can be thought of as an approximate inverse of $G^c$ and $G^s$, respectively. If $\mathfrak{Z}^c$ and $\mathfrak{Z}^s$ are invertible, then we can transform the random variables exactly. Otherwise, we can learn an estimate of $p(\mathbf{z}^c|\mathbf{x}, \mathbf{y}, i)$ and $p(\mathbf{z}^s|\mathbf{x}, i)$ by knowing $\phi^c(\mathbf{x})$ and $\phi^s(\mathbf{x})$. In Appendix C.2.3, we show how we can estimate $\mathfrak{Z}^c$ and $\mathfrak{Z}^s$ with simple models in an unsupervised manner. One benefit of this is transferring the content

of an image from one client's domain to another, which includes anonymization as a special case. We can look at the problem of inferring $\mathbf{z}^c, \mathbf{z}^s$ given $\mathbf{x}, \mathbf{y}$ from client $i$ as a problem of "inverting the data generating process" (Zimmermann et al., 2021). We will next show how we can encourage disentanglement between $\mathbf{z}^c$ and $\mathbf{z}^s$ using a contrastive regularizer on the discriminators.

Before we move on, note that due to federated averaging, we have $\theta_{i,\mathbf{z}^c}^{t+1,0} = \mathbb{E}_{p(i)}\theta_{i,\mathbf{z}^c}^{t,\tau}$, where $\theta_{i,\mathbf{z}^c}^{t,\tau}$ is the parameter of $G_i^c$ in client $i$ and $\tau$ is the number of local steps. This applies to all aggregated parts of the model. Thus, assuming that a global optimal solution $\theta_{\mathbf{z}^c}^*$ exists, we have $p(\mathbf{z}^c|\mathbf{x}, i; \theta_{i,\mathbf{z}^c}^{t,0}) \xrightarrow{t\to\infty} p(\mathbf{z}^c|\mathbf{x}; \theta_{\mathbf{z}^c}^*)$, and the same goes for all federated parameters.

### 4.3 Contrastive Regularization

The main motivation for partitioning the model into federated and private components is that the federated components should be biased towards a solution that is invariant to the client's domain $i$. This is a desirable property to have for predicting $\mathbf{y}$ given $\mathbf{x}$. However, from a generative perspective, we still want to learn $\mathbf{x}$ given $i$ and $\mathbf{y}$, so we are interested in learning a personalized generative model that can model the difference in changing only the client $i$ while keeping $\mathbf{x}$ and $\mathbf{y}$ fixed, and similarly changing only $x$ while keeping $i$ and $\mathbf{y}$ fixed. This type of intervention can help us understand how to represent these variations better.

Suppose that we do not know $\mathbf{y}$. We want to learn how $i$ influences the content in $\mathbf{x}$. The variable $i$ might have variations that correlate with the content, but the variations that persist across multiple clients should be the ones that can potentially identify the content of the data. From another point of view, data with similar content might still have a lot of variations within one client. Still, if these variations are uninformative according to the other clients, then we should not regard them as content-identifying variations. Therefore, we propose to maximize the correlation between the content representation of two generated images given the same content latent $\mathbf{z}^c$. This is also applied analogously to style. To achieve that, we use the Barlow Twins objective (8) as a regularizer and the generator as a model for simulating these interventions. Indeed, such interventional data are sometimes impossible to sample on demand from nature, so simulating an intervention on a good enough generative model can help as a surrogate to reality. For example, we can ask what the digit 5 would look like, what could be considered a chair, or what could be classified as a disease or a tumor according to client A vs. B.

Let $g$ be a projector from the feature space to some metric embedding space. With some abuse of notation, let $\# \in \{c, s\}$ denote the component we are interested in. Given a generative sample $\tilde{\mathbf{x}} \sim p(\mathbf{x}|\mathbf{z}^c, \mathbf{z}^s, i)$ let us denote the embedding of $\phi_i^{\#}(\tilde{\mathbf{x}})$ as $\Phi_i^{\#}(\mathbf{z}^c, \mathbf{z}^s) = g_i\left(\phi_i^{\#}(\tilde{\mathbf{x}})\right)$, where we compute $\tilde{\mathbf{x}} = \text{sg}\left(G_i^c(\mathbf{z}^c, G_i^s(\mathbf{z}^s))\right)$. Note the usage of the stop gradient operator 'sg' to emphasize that $\tilde{\mathbf{x}}$ is treated as a sample from $p$, so we do not pass gradients through it. Given this notation, the Barlow Twins regularizer we propose can be written as follows

$$\Gamma_i(\theta; \mathbf{z}^c, \mathbf{z}^s) = \underset{p(\tilde{\mathbf{z}}^c, \tilde{\mathbf{z}}^s|i)}{\mathbb{E}} [\mathcal{L}_{\text{BT}}\left(\Phi_i^c(\mathbf{z}^c, \mathbf{z}^s), \Phi_i^c(\mathbf{z}^c, \tilde{\mathbf{z}}^s)\right) + \mathcal{L}_{\text{BT}}\left(\Phi_i^s(\mathbf{z}^c, \mathbf{z}^s), \Phi_i^s(\tilde{\mathbf{z}}^c, \mathbf{z}^s)\right)]. \tag{14}$$

The intuition here is that the correlation between the content features of a pair of samples generated by fixing the content latent and changing other latent should be close, and similarly for the style. Assuming optimal generator, we can simulate real-world interventions and self-supervise the model via this contrastive regularization. Note that the term "contrastive" is not concerned with positive and negative samples as is common in the self-supervised learning literature. Instead, we contrast a pair of samples that are the same except for some latent variable, so it could be understood as being "latent-contrastive".

If we let $\mathcal{L}_{\text{BT}}^{\#}(\mathbf{x}_1, \mathbf{x}_2) = \mathcal{L}_{\text{BT}}\left(g_i\left(\phi_i^{\#}(\mathbf{x}_1)\right), g_i\left(\phi_i^{\#}(\mathbf{x}_2)\right)\right)$, then we can rewrite the objective in terms of interventions on the generator's latent variables

$$\Gamma_i(\theta; \mathbf{z}^c, \mathbf{z}^s) = \underset{\substack{\tilde{\mathbf{z}}^s \sim \\ p_i(\mathbf{z}^s)}}{\mathbb{E}} \left[ \underset{\substack{\mathbf{x}_1, \mathbf{x}_2 \sim \\ p_i(\mathbf{x}|\tilde{\mathbf{z}}^s, do(\mathbf{z}^c))}}{\mathbb{E}} \mathcal{L}_{\text{BT}}^c(\mathbf{x}_1, \mathbf{x}_2) \right] + \underset{\substack{\tilde{\mathbf{z}}^c \sim \\ p_i(\mathbf{z}^c)}}{\mathbb{E}} \left[ \underset{\substack{\mathbf{x}_1, \mathbf{x}_2 \sim \\ p_i(\mathbf{x}|\tilde{\mathbf{z}}^c, do(\mathbf{z}^s))}}{\mathbb{E}} \mathcal{L}_{\text{BT}}^s(\mathbf{x}_1, \mathbf{x}_2) \right],$$

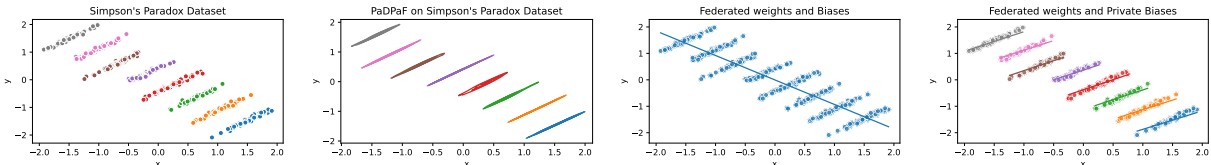

Figure 3: Our partially-federated linear generator can learn the right data-generating process for this dataset (top right). Federated averaging on a linear model will yield the ordinary least squares solution at best (bottom left), whereas a linear model with a private bias has better generalization per client (bottom right). The top left plot shows the full dataset. (color = client)

Finally, for some hyperparameter $\lambda$, we can rewrite the objective (1) with our regularizer

$$F_i(\theta; \lambda) = \mathbb{E}_{\mathbf{x},\mathbf{y}} \left[ f_i(\theta; \mathbf{x}, \mathbf{y}) \right] + \lambda \mathbb{E}_{\mathbf{z}^c, \mathbf{z}^s} \Gamma_i(\theta; \mathbf{z}^c, \mathbf{z}^s). \tag{15}$$

See Fig. 2 for a better understanding of the interventions.

**Implementation details.** The algorithm for training our model is a straightforward implementation of GAN training in a federated learning setting. We use `FedAvg` (McMahan et al., 2016) algorithm as a backbone. The main novelty stems from combining a GAN architecture that can be decomposed into federated and private components, as depicted in Fig. 1, with a latent-contrastive regularizer (14) for the discriminator in the GAN objective. The training algorithm and other procedures are described in more detail in the Appendix.

## 5 Experiments

In this section, we show some experiments to show the capabilities of the PaDPaF model. First, we run a simplified version of the PaDPaF model on a simple linear regression problem with data generated following Simpson's Paradox as shown in Fig. 3. Next, we run the main experiment on MNIST (LeCun et al., 1998) and CIFAR-10 (Krizhevsky, 2009). We compare our method with Ditto (Li et al., 2021a) as well as Ditto with FedProx (Li et al., 2018). We further demonstrate that our method also works with variational auto-encoders (VAEs) (Kingma & Welling, 2013). Finally, we show our model's performance on CelebA (Liu et al., 2015), mainly to show its abilities in generating and varying locally available attributes (i.e. styles) on data sharing the same content. More experimental details can be found in the supplementary material. We generally use Adam (Kingma & Ba, 2014) for both the client's and the server's optimizer.

In our experiments, we show how our generator disentangles content from style visually. We also show how the generator can generate locally unseen labels or attributes, i.e., generate samples from the client's distribution given labels or attributes that the client has never seen. Next, we show how the features learned (without supervision) by the content discriminator contain sufficient information such that a simple classifier on the features can yield high accuracy on downstream tasks.

**Linear Regression.** We generate a federated dataset with Simpson's Paradox by fixing a weight $w$ and changing the bias $b$ across clients $i$, which would be $y = wx + b_i$. Further, for each client, we sample $x$ from intervals such that the opposite trend is observed when looking at data from all clients (see Fig. 3).

Training a naive federated linear classifier leads to seemingly acceptable performance but learns the opposite true trend, as the negative trend is only due to sampling bias $x|i$. By federating the weights and keeping the biases private, we allow each client to learn a shared trend and locally learn their own biases. Then, given that each client has a discriminator that can tell with high accuracy whether some data is sampled from their distributions or not, we can achieve significantly smaller generalization error by routing new data to the most confident client and taking its prediction. Alternatively, we can weight the predictions of the clients based on their discriminators' probabilities (i.e. compute $p(\hat{\mathbf{y}}(x)) := \mathbb{E}_{p(\mathbf{x}=x,i)} p(\mathbf{y}|\mathbf{x} = x, i)$ vs. $p(\hat{\mathbf{y}}(x)) := p(\mathbf{y}|\mathbf{x} = x, i = i')$, where $i' := \arg\max_i p(\mathbf{x}|i)$). Given good enough discriminators and expressive enough generators, we can generalize better while giving the clients the flexibility to handle the private parameters completely on their own to their benefit.

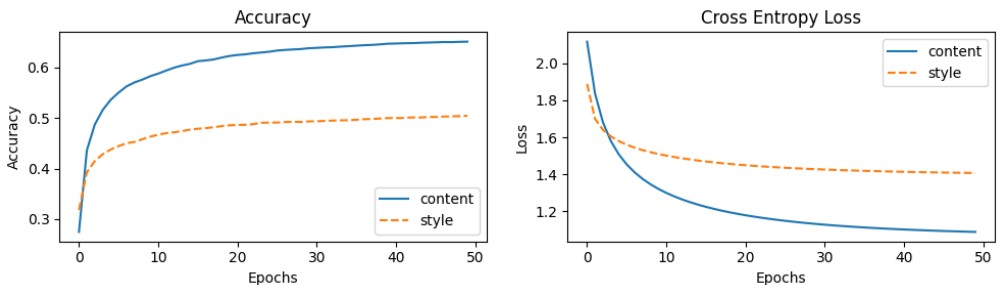

Figure 4: Left two plots: Accuracy and loss of a linear classifier on the content and style features of unsupervised GAN training on MNIST (i.e. labels **y** are not used in training). Right two plots: The progress of the accuracy and loss achieved by a linear classifier in terms of communication rounds (better seen with colors).

Figure 5: Accuracy and loss of a linear classifier on the content and style features of unsupervised GAN training on CIFAR-10 (showing performance on the style features of one client only)

.

The model we consider for this problem is a simple linear model for the generator with federated weight and private bias, and a small MLP for the discriminators (as it is impossible to linearly separate the clients in the middle from the rest). For all clients $i$, we generate a point $\mathbf{x} = w\mathbf{z}^c + b_i$, where $\mathbf{z}^c \sim \mathcal{N}(0,1) \in \mathbb{R}$ and $\mathbf{x}, w, b_i \in \mathbb{R}^2$, where we choose $w$ to be federated and $b_i$ to be private. However, we can also consider an agnostic additive parameterization between federated and private parameters as $\mathbf{x} = w^c\mathbf{z}^c + w^s_i\mathbf{z}^s + b^c + b^s_i$, where $\mathbf{z}^s \sim \mathcal{N}(0,1)$, $w^c$ and $b^c$ are federated, and $w^c$ and $b^c$ are private. We show the performance of the latter model in Fig. 3.

**MNIST.** In our MNIST experiments, we generate a federated dataset by partitioning MNIST equally on all clients and then adding a different data augmentation for each client. This way, we can test whether our model can generate global instances unseen to some clients based on their data augmentation. The data augmentations in the figures are as follows (in order): 1) zoom in, 2) zoom out, 3) color inversion, 4) Gaussian blur, 5) horizontal flip, 6) vertical flip, 7) rotation of at most 40 degrees, and 8) brightness jitter.

See Fig. 6 for samples from a conditional GAN (i.e. labels **y** are used in training). One interesting observation about the samples is that, since the number 7 was only seen in the horizontally flipped client, it retains this flipped direction across other clients as well. This is because learning flipping as a style is not incorporated in the design of our generator, and the same goes for rotations (see Fig. 6, lower left). However, the content was generally preserved throughout other clients. We argue that better architectures can mitigate this issue. Performance on other more difficult augmentations can be seen in the Appendix.

From Fig. 4, we see the content features contain sufficient information such that a linear classifier can achieve 98.1% accuracy on MNIST without supervision. It is also clear that the style features do not contain as much information about the content. This empirically confirms that our method can capture the most from $\mathbf{z}^c$ and disentangle it from $\mathbf{z}^s$.

In comparison with the baselines Ditto and Ditto+FedProx, it is clear by visual inspection from Fig. 7 that our method is better and faster. We corroborate this by reporting the FID scores as well Seitzer (2020) for each method in Tab. 1. Furthermore, the results from the VAE experiment in Fig. 8 show that our idea can extend to other generative models as well, which is a direction for future work.

| Alg. / id. | #1 | #2 | #3 | #4 | #5 | #6 | #7 | #8 | Average |
|---|---|---|---|---|---|---|---|---|---|
| PaDPaF | 11.47 | 4.10 | 11.12 | 7.77 | 8.38 | 6.84 | 7.94 | 8.62 | 8.28 |
| Ditto | 29.70 | 86.13 | 240.70 | 39.75 | 22.03 | 21.34 | 21.43 | 22.61 | 60.46 |
| Ditto+FedProx | 90.87 | 110.90 | 170.13 | 49.71 | 63.02 | 54.01 | 58.55 | 53.545 | 53.55 |

Table 1: FID for each client's GAN model and personalized training algorithm.

Figure 6: Samples from a conditional GAN on MNIST. Top to bottom, $\mathbf{z}^c$ is changed while $\mathbf{z}^s$ and $\mathbf{y}$ are fixed. Left to right, $\mathbf{z}^c$ is fixed while $\mathbf{z}^s$ and $\mathbf{y}$ are changed. $\mathbf{z}^c$ and $\mathbf{z}^s$ are the same across clients.

**CIFAR-10.** We prepare the clients' data in the same way as the MNIST experiments but with slightly different data augmentations and 10 clients instead. The data augmentations are as follows (in order): 1) zoom in, 2) color inversion, 3) Gaussian blur, 4) grayscale, 5) vertical flip, 6) brightness jitter, 7) random perspective (affine), and 8) solarize, 9) custom color transform 1, and 10) custom color transform 2. The model demonstrates the same benefits from the MNIST experiments, as can be seen from Fig. 5. The content features learned correspond to the content more than the style features. We show some samples from the model in Appendix C.3 as well.

**CelebA.** The federated dataset is created by a partition based on attributes. Given 40 attributes and 10 clients, each client is given 4 unique attributes and only has access to data having any of these 4 attributes. The content part should then capture the general facial structures, and the style should capture the other local variations within the attributes. Though not state-of-the-art in terms of image fidelity, our results confirm the empirical feasibility of our model. The generated data for the client has a variety that is specific to the attributes available to the client.

# 6 Discussion

Our framework leaves some room for improvements and further work to be done in multiple directions. Here, we discuss some of those directions and highlight interesting aspects we can exploit.

**Client shift.** Our architecture adapts to different clients $i$ under covariate shift $q_i(\mathbf{x})$ and prior shift $q_i(\mathbf{y})$. Can we remove assumption Assumption 1 and extend our framework to the case of conditional shift $q_i(\mathbf{y}|\mathbf{x})$? This would help personalize the model's prediction based on the client.

**Model improvement.** How can we design a generator that can effectively incorporate the clients' variations? Is the choice of private parameters mostly problem-specific? What about different modalities of data, such as language and time series?

**Generator on the server.** Our model is trained with `FedAvg` and can benefit from the same privacy guarantees. Privacy in our case could be better because the federated part of the generator can stay completely in the server and the discriminator can share only the federated part of its parameters. Can we train the federated discriminator completely on the server as well to ensure complete privacy? The server would then need data samples or at least discriminative features of the data from the clients, which might reveal private information about the client. Still, it would be interesting to explore the benefits of having the generator be completely on the server Augenstein et al. (2020).

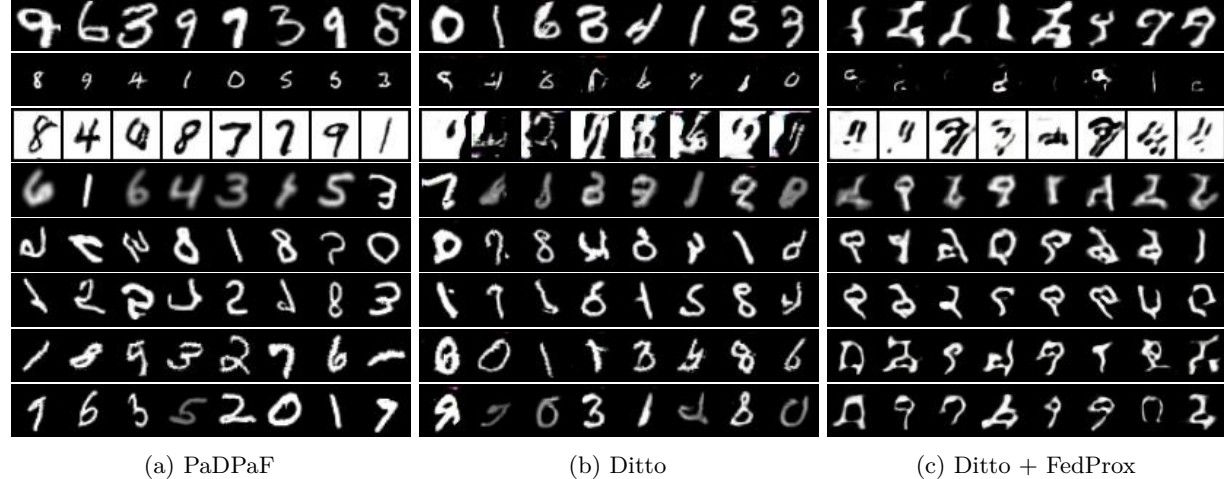

| (a) PaDPaF | (b) Ditto | (c) Ditto + FedProx |

Figure 7: Samples from clients trained for 120 rounds with different personalization methods.

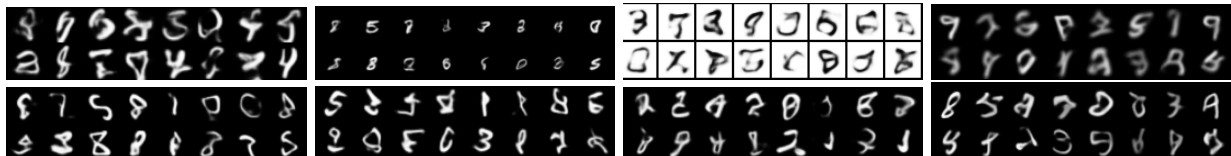

Figure 8: VAE samples after 100 rounds of training with PaDPaF.

**Conclusion.** We introduced a framework combining federated learning, generative adversarial learning, and (causal) representation learning together. We do so by leveraging the federated setting and learning a client-invariant representation based on a causal model of the data-generating process, which we model as a GAN. We disentangle the content and style latents using a contrastive regularizer on the discriminators. Experiments validate our framework's benefits and show that our model is effective at personalized generation and client-invariant classification, and benefits further from supervised data.

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

## A  Algorithm

In this section, we write the training algorithm in detail for completeness. The training is a straightforward application of `FedAvg` on GANs without its private parameters.

Here, we let $\text{OPT}(\theta, \nabla f(\theta))$ be a first-order optimizer, such as SGD or Adam, that takes the current parameters $\theta$, and the gradient $\nabla f(\theta)$ and returns the updated parameters given step size $\eta$. We also define a subroutine $\text{GANOPT}$ that runs a single descent or ascent step on the GAN's objective. The term $\text{train}_G$ is a boolean variable that specifies whether the generator or the discriminator should be trained. In the GAN literature, the discriminator is often trained 3 to 5 times as much as the generator. In our experiments, we let $T_D = 3$, so $D$ is trained 3 times before $G$ is trained, but we also update $\theta^D$ with double the learning rate, following the Two Time-scale Update Rule (Heusel et al., 2018). As for the client's loop, we can simply train it on the dataset for one epoch before breaking the loop so that $t_{\max} = |X_i|$ and $\pi_i = 0$, but we add a loopless option as well by letting $t_{\max} = \infty$ and $\pi_i \propto |X_i|$ for the same training behavior on average.

Note that we could have removed line 12 of the algorithm and let line 11 be $\Delta_i^{\text{fed}} \leftarrow \theta_i^{\text{fed}}(1) - \theta_i^{\text{fed}}(t)$, but we chose the current procedure to emphasize that the server's update on the private parameters is always set to 0.

## B  Model Architecture

Our model's architecture is composed of regular modules that are widely used in the GAN literature. The generator is a style-based generator with a ResNet-based architecture (Gulrajani et al., 2017), where we use a conditional batch normalization layer (De Vries et al., 2017) to add the style, which is generated via a style vectorizer (Karras et al., 2019). The discriminator's architecture is ResNet-based as well (He et al., 2016) with the addition of spectral normalization (Miyato et al., 2018). In the case of conditioning on labels, we follow the construction of the projection discriminator as in (Miyato & Koyama, 2018), so we add an embedding layer to the discriminator, and we adjust the conditional batch normalization layers in the generator by separating the batch into two groups channel-wise, and condition each group separately. Finally, we add a projector to the discriminator for contrastive regularization, which is a single hidden layer ReLU network. Recall that we use two discriminators, one is private, and the other is federated.

As for the model architecture for the linear regression dataset, we simplify the discriminator to a 2-layer neural net with 8 hidden dimensions. The generator is a linear layer and the style vectorizer is also a linear layer with a latent vector of dimension 1. The projectors are linear layers as well.

The variables that control the size of our networks are the image size, the discriminator's feature dimension, and the latent variables' dimension. We make the dimension of the content and style latent variables equal for simplicity. For MNIST, we choose the feature dimension to be 64 and the latent dimension to be 128. For CelebA, we choose the feature dimension to be 128 and the latent dimension to be 256. Kindly refer to the code for more specific details.

---

**Algorithm 1** Contrastive GANs with Partial FedAvg

---

1: **Input:** Clients $i \in [M]$, datasets $X_i = \{(\mathbf{x}_j, \mathbf{y}_j)\}_{j=1}^{|X_i|}$, target distribution $q(\cdot|i)q(i)$ over $X_i$, GAN models $p_i(\cdot; \theta_i)$, GAN objectives $f_i$, server and clients first-order optimizers $\text{OPT}_0$ and $\text{OPT}_i$ with step sizes $\eta_0$ and $\eta_i$, client's loop stopping probability $\pi_i \propto |X_i|^{-1}$, and $T_D$ iterations for training $D$ vs. $G$.

2: **Output:** $\theta_0^{\text{fed}}$ and $\theta_i^{\text{pvt}}, \forall i$.

3: Set $\theta_i := \theta_0, \forall i \in [M]$

4: **for** $\tau = 1, \cdots, \tau_{\max}$ **do**

5:      Sample subset of clients $I \sim q(i)$

6:      **for** $i \in I$ **in parallel do**

7:          **for** $t = 1, \cdots, t_{\max}$ **do**

8:              $\text{train}_G \leftarrow \mathbb{1}[t \bmod T_D + 1 = 0]$

9:              $\theta_i(t+1) \leftarrow \text{GANOPT}(i, \theta_i(t), \text{train}_G; f_i, q_i)$

10:             **if** $1 \sim \text{Bernoulli}(\pi_i)$ **then**

11:                 $\Delta_i \leftarrow \theta_i(1) - \theta_i(t)$

12:                 $\Delta_i^{\text{pvt}} \leftarrow 0$

13:                 **break loop**

14:             **end if**

15:          **end for**

16:      **end for**

17:      $\theta_0(\tau+1) \leftarrow \text{OPT}_0\left(\frac{1}{|I|} \sum_{i \in I} \Delta_i; \theta_0(\tau), \eta_0\right)$

18:      $\theta_i^{\text{fed}} \leftarrow \theta_0^{\text{fed}}$ for all clients $i \in [M]$

19: **end for**

20: **procedure** $\text{GANOPT}(i, \theta_i, \text{train}_G; f_i, q_i)$

21:      **if** $\text{train}_G$ **then**

22:          **return** $\text{OPT}_i\left(\nabla_{\theta_i^G} f_i(\theta_i, \cdot); \theta_i, \eta_i\right)$

23:      **else**

24:          Sample $\xi \sim q(\cdot|i)$

25:          **return** $\text{OPT}_i\left(-\nabla_{\theta_i^D} f_i(\theta_i, \xi); \theta_i, 2\eta_i\right)$

26:      **end if**

27: **end procedure**

---

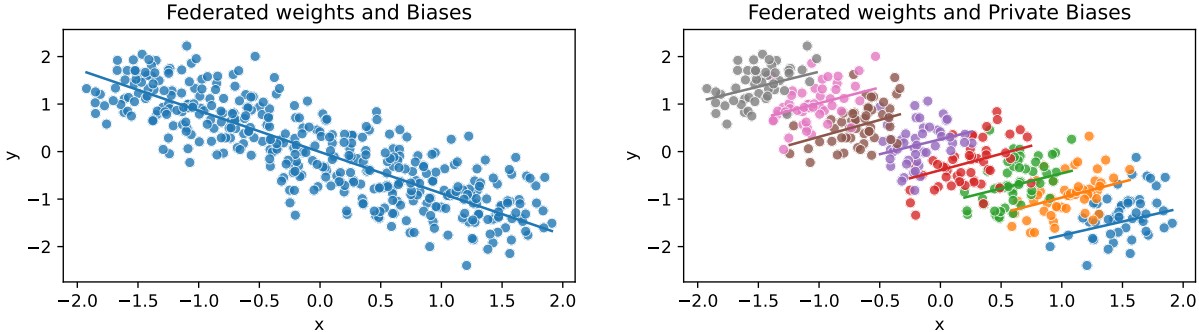

Figure 9: A single linear regression model on Simpson's Paradox dataset vs. federated linear regression models with private biases.

## C  Experiments

In this section, we describe the experiments we run in more detail and show results extra results supporting our framework.

For all our experiments, the client models are handled by "workers". The workers only train their models on a specific subset of clients that does not change. We often assign a worker for each client, but due to constraints on computational resources, we might create a smaller number of workers and assign a specific subset of clients for each worker. We run all experiments on a single NVIDIA A100 SXM GPU 40GB.

Our choice for all the optimizers, including the server's optimizer, is Adam (Kingma & Ba, 2014) with $\beta_1 = 0.5$ and $\beta_2 = 0.9$. We choose Adam due to its robustness and speed for training GANs. We found that choosing learning rates 0.01 and 0.001 for the server optimizer and the client optimizer, respectively, is a good starting point. We also use an exponential-decay learning rate schedule for both the server and the clients, with a decay rate of 0.99 for MNIST (0.98 for CelebA) per communication round. This can help stabilize the GAN's output. For Ditto and FedProx, we choose the prox parameter to be equal to 1.0.

### C.1  Linear Regression on Simpson's Paradox

We generate a dataset that demonstrates Simpson's Paradox for $M$ clients as follows. We fix a weight, say, $w := 1$. Then for each client $i \in \{1, \cdots, M\}$, we sample $\mathbf{x}$ uniformly from $[M - i, M - i + 1]$ and a bias $b_i$ from $[L(i - 1), Li]$ for some constant $L$. Finally, for each client $i$, we generate $n_i$ points $y \sim w\mathbf{x} + b_i$ and then normalize $\mathbf{x}*$ and $\mathbf{y}$ by subtracting the mean and dividing by the standard deviation. In our experiments, we choose $M := 8$, $L := 4$, and $n_i := 50$ for all $i$. For training a PaDPaF model on this dataset, we noticed that $D^c$ should have a slightly larger feature dimension, like 16, whereas $D_i^s$ can be smaller, like 2.

If we train a naive linear regression model on this dataset for 50 epochs with a batch size of 10, we would get a mean-squared error of about 0.182, whereas similarly training a linear regression model for each client, and then freezing the models and training again a linear routing model would yield an error about 0.004, which is two orders of magnitude smaller (see Fig. 3). In Fig. 9, we see that we can still find the right trend for each client with more cluttered data.

Code for reproducing plots and similar errors for this experiment are provided in the linked repository in the main text.

### C.2  MNIST

The MNIST train dataset is partitioned into 8 subsets assigned to 8 clients, and each client is handled by a unique worker. The data augmentations used for each partition are clear from the images. We write the PyTorch code for each data augmentation:

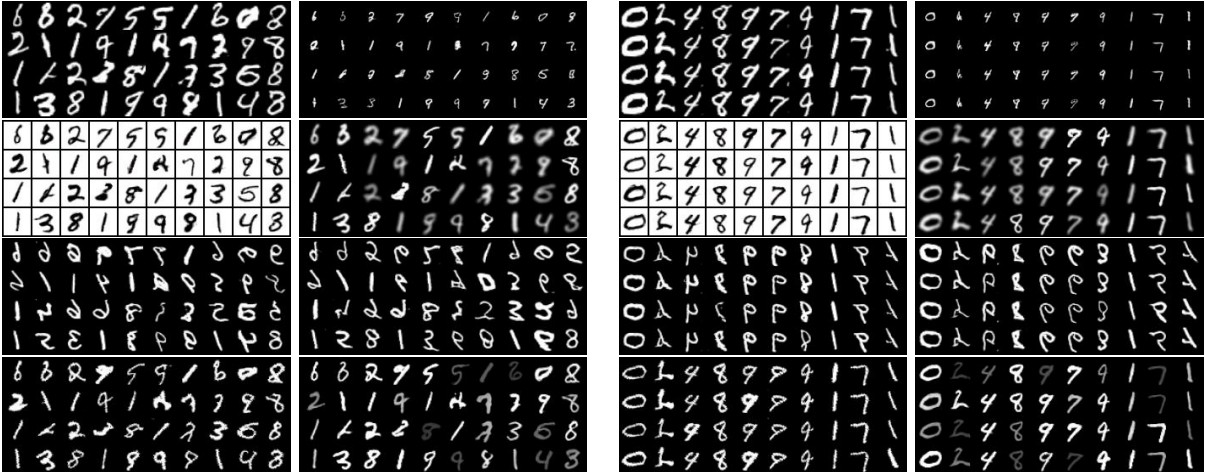

(a) $\mathbf{z}^c$ and $\mathbf{z}^s$ are the same across clients, but different within clients.

(b) Top to bottom, $\mathbf{z}^s$ is changed while $\mathbf{z}^c$ and $\mathbf{y}$ are fixed. Left to right, $\mathbf{z}^s$ is fixed while $\mathbf{z}^c$ and $\mathbf{y}$ are changed.

Figure 10: Samples from a GAN without conditioning. $\mathbf{z}^c$ and $\mathbf{z}^s$ are consistent across clients.

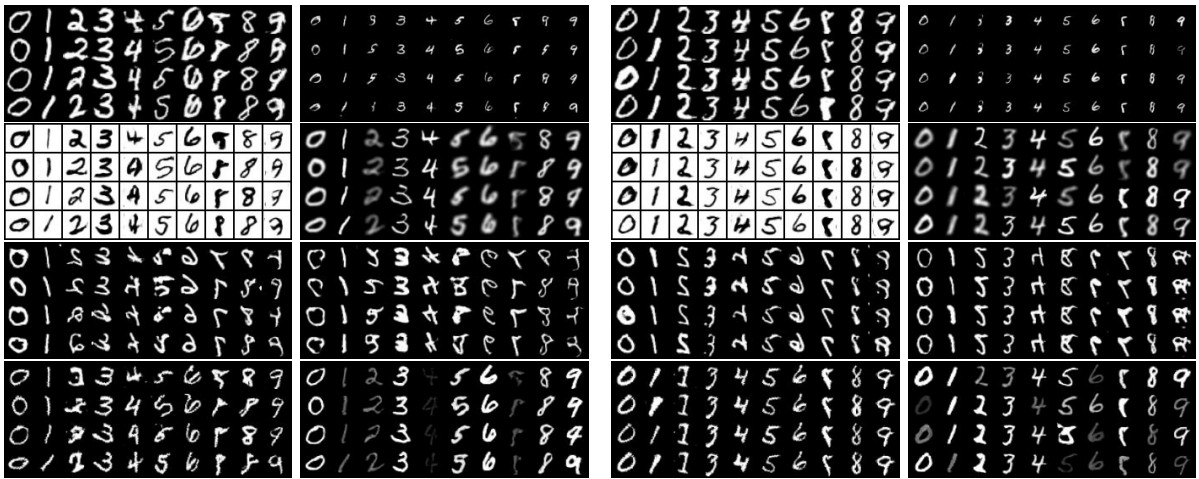

(a) Top to bottom, $\mathbf{z}^c$ is changed while $\mathbf{z}^s$ and $\mathbf{y}$ are fixed. Left to right, $\mathbf{z}^c$ is fixed while $\mathbf{z}^s$ and $\mathbf{y}$ are changed.

(b) Top to bottom, $\mathbf{z}^s$ is changed while $\mathbf{z}^c$ and $\mathbf{y}$ are fixed. Left to right, $\mathbf{z}^s$ is fixed while $\mathbf{z}^c$ and $\mathbf{y}$ are changed.

Figure 11: Samples from a conditional GAN. $\mathbf{z}^c$ and $\mathbf{z}^s$ are consistent across clients.

1. Zoom in: `CenterCrop(22)`.

2. Zoom out: `Pad(14)`.

3. Color inversion: `RandomInvert(p=1.0)`.

4. Blur: `GaussianBlur(5, sigma=(0.1, 2.0))`.

5. Horizontal flip: `RandomHorizontalFlip(p=1.0)`.

6. Vertical flip: `RandomVerticalFlip(p=1.0)`.

7. Rotation: `RandomRotation(40)`.

8. Brightness: `ColorJitter(brightness=0.8)`.

In the case of conditioning on **y**, we further restrict the datasets and drop 50% of the labels **from each partitioned dataset**. This implies that some data will be lost, and each client will see only 50% of the labels. This is intentional as we want to restrict the dataset's size further and test our model's robustness on unseen labels.

The results for training our model on this federated dataset are shown in Fig. 10 for the unconditional case (i.e. **y** is unavailable), and Fig. 11 for the conditional case with 50% labels seen per client. The results were generated after training the model for 300 communication rounds. For MNIST, we train the models for a half epoch in each round to further restrict the local convergence for each client.

### C.2.1 Adaptation to New Clients (i.e. Data Augmentation)

After training our conditional model on the previous data augmentations, we re-train it for 25 communication rounds on the following data augmentations and show the results:

1. Affine: `RandomAffine(degrees=(30, 70), translate=(0.1, 0.3), scale=(0.5, 0.75))`.

2. Solarize: `RandomSolarize(threshold=192.0)`.

3. Erase: `transforms.RandomErasing(p=1.0, scale=(0.02, 0.1))` (after `ToTensor()`).

This is to show that our conditional model can quickly adapt to new styles without severely affecting its content generation capabilities and its generalization to unseen labels. We can see in Fig. 13 that solarization is learned as a style, while erosion is learned as a content. The generalization to unseen digits is maintained but less so in the challenging affine augmentation.

### C.2.2 Can We Predict the Client from the Image?

We ask a question analogous to the one we are concerned with in representation learning. Instead of predicting the label from the image, we want to predict the augmentation, or the client, that generated the image. We show that we can predict the client with good accuracy (93.3%). See Fig. 14.

The style prediction is done by linear transformations on the content representation and the style representations. For the style representations, we linearly map each representation from each client to a scalar $u_i$, so that $u$ is a vector of all the scalars. We then pass $u$ through an extra linear transformation to get a vector of logits for $p(i|\mathbf{x})$. Note that this is still a linear transformation of the style representations, but we do it this way to reduce dimensionality.

Note that the content representation can still predict the style with good accuracy. Indeed, we have seen that our generator architecture finds difficulties in assigning rotations to style variations, which is due to the style generator construction as a conditional batch normalization layer, which shows limited capabilities in capturing rotation-like variations. Still, the prediction accuracy from the style representations is better overall, which shows that our model does disentangle, to some extent, the style from the content.

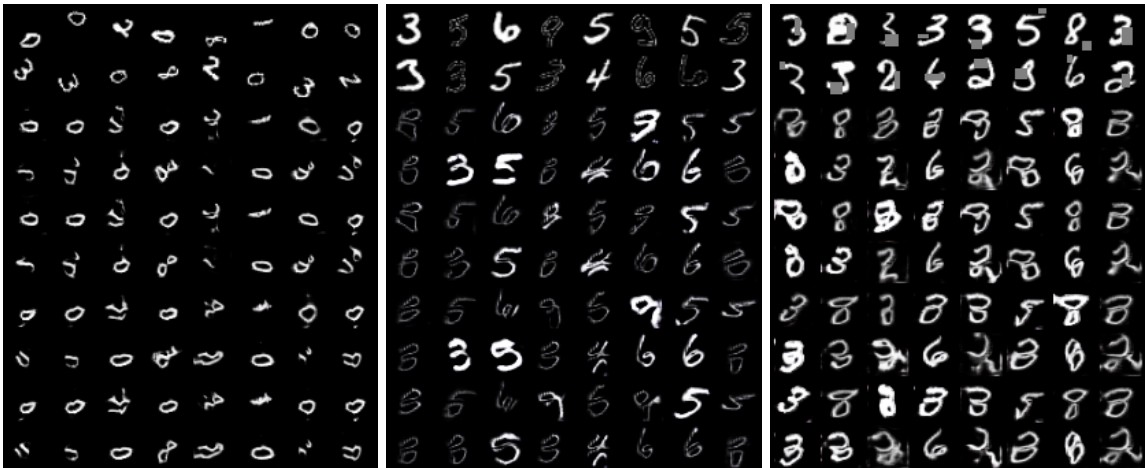

Figure 12: Results after re-training for 1 communication round on the three new data augmentations. Rows 1-2 are real samples and rows 3-8 are generated. Rows 3-6 share the same $\mathbf{z}^c$, and rows 7-10 as well. Rows 3-4 and 7-8 share the same $\mathbf{z}^s$, similarly for rows 5-6 and 9-10.

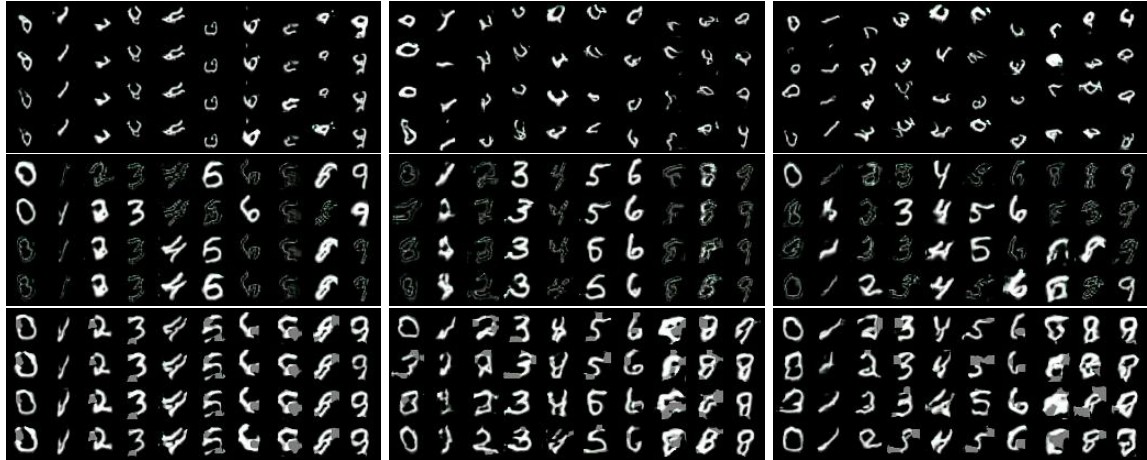

(a) Changing $\mathbf{z}^s$ per row, fixing $\mathbf{z}^c$. (b) Changing $\mathbf{z}^c$ per row, fixing $\mathbf{z}^s$. (c) Changing both $\mathbf{z}^c$ and $\mathbf{z}^s$.

Figure 13: Results on unseen clients after 25 communication rounds.

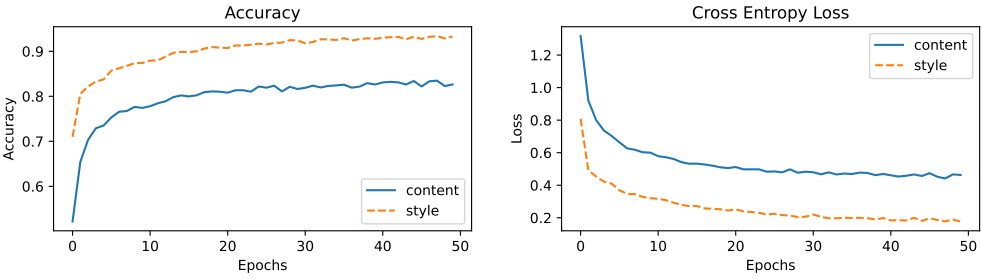

Figure 14: Prediction of style. We see that the content features have good accuracy (82.6%), but the style features can classify the source client with much better accuracy (93.3%).

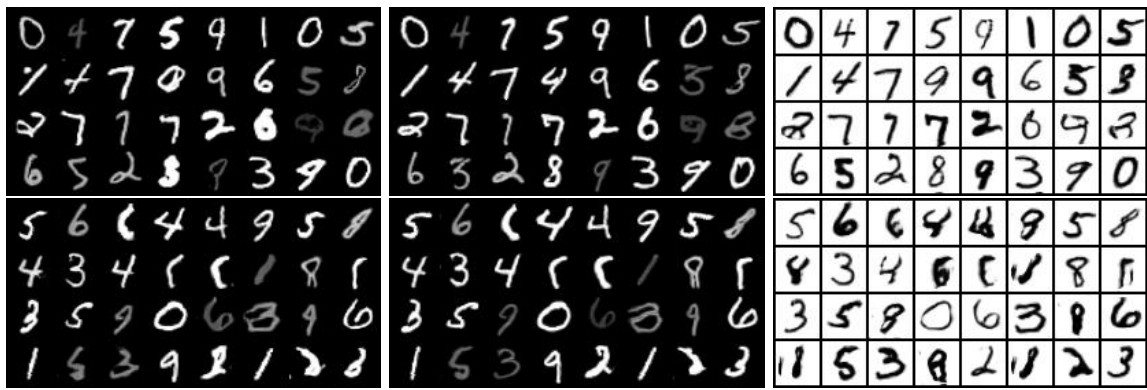

(a) Source (generated) images.        (b) Reconstructed images.        (c) Re-styled, transferred content.

Figure 15: The images show results from the regular, unconditional GANs, whereas the bottom shows results for conditional GANs with 50% labels seen per client. Observe how the content stays preserved after transfer. Here, we show an example of transferring the content from client 8 (brightness) to client 3 (color inversion).

### C.2.3   Re-Styling via Content Transfer

We describe a simple method for transferring the content of an image to another style, given the style vectorizers $G_i^s$. In other words, we can transfer the content of an image taken from one client to another without sharing the image but by finding the approximate latent variables that generate the image and transferring the content latent variable. We shall see that this approach can indeed transfer the content well enough, where the content is up to the generator's expressiveness and its separation of style. For example, we show that our generator finds difficulties in distinguishing some rotations as non-content variations. This is because all other clients have slightly rotated digits as well.

To find the approximate latent variables from the image without any significant overhead, we use the discriminator's representations to predict the latent variables. We show that this approach can predict latents that reconstruct the image very well, and thus we can use the same content latent on the new client.

---

**Algorithm 2** Re-style via content transfer

---

1: **Input:** Trained models, $t_{\max}$ number of epochs, $\mathbf{x}_a$ (and $\mathbf{y}_a$, if any) a sample from source client $a$, and a target client $b$.
2: **Output:** Representation-to-latent transforms $\mathfrak{Z}^c$ and $\mathfrak{Z}_i^s$ for each $D_0^c$ and $D_i^s$, resp., $\forall i \in [M]$, and re-styled $\mathbf{x}_b$.
3: **for** $t = 1, \cdots, t_{\max}$ **do**
4:     Sample client $i \sim q(i)$, and sample label $\mathbf{y} \sim q_i(\mathbf{y})$, if any.
5:     Sample latents $\mathbf{z}^c \sim q_i(\mathbf{z}^c)$ and $\mathbf{z}^s \sim q_i(\mathbf{z}^s)$.
6:     Sample image $\mathbf{x} \sim p_i(\mathbf{x}|\mathbf{z}^c, \mathbf{z}^s, \mathbf{y})$ (i.e. set $\mathbf{x} = G_0^c(\mathbf{z}^c, G_i^s(\mathbf{z}^s), \mathbf{y})$).
7:     Transform latents $\tilde{\mathbf{z}}^c = \mathfrak{Z}^c(\phi_0^c(\mathbf{x}))$ and $\tilde{\mathbf{z}}_i^s = \mathfrak{Z}_i^s(\phi_i^s(\mathbf{x}))$.
8:     Sample reconstructed image $\tilde{\mathbf{x}} \sim p_i(\mathbf{x}|\tilde{\mathbf{z}}^c, \tilde{\mathbf{z}}^s, \mathbf{y})$.
9:     Minimize $\mathcal{L}_{\mathrm{BT}}(\phi_0^c(\mathbf{x}), \phi_0^c(\tilde{\mathbf{x}}); g_0^c)$ w.r.t. $\mathfrak{Z}^c$ and $\mathfrak{Z}_i^s$ (see (8)).
10: **end for**
11: **Transfer** $a \to b$**:** Sample $\mathbf{x}_b \sim p_b(\mathbf{x}|\mathfrak{Z}_0^c(\phi_0^c(\mathbf{x}_a)), \mathbf{z}_b^s, \mathbf{y}_a)$, where $\mathbf{z}_b^s \sim p_b(\mathbf{z}^s)$.

---

First, we show the algorithm for reconstructing and re-styling an image from some client. The main idea is to use $\mathcal{L}_{\mathrm{BT}}$ from (8) to maintain an identity correlation between the source content representation and the target (i.e. reconstructed) content representation. We noticed that we do not need to do the same for the style representation. Not enforcing an identity correlation between styles can even improve the results in terms of style consistency. We do not find a good explanation for this. We believe that applying Barlow Twins loss in our setting warrants further empirical and theoretical investigation.

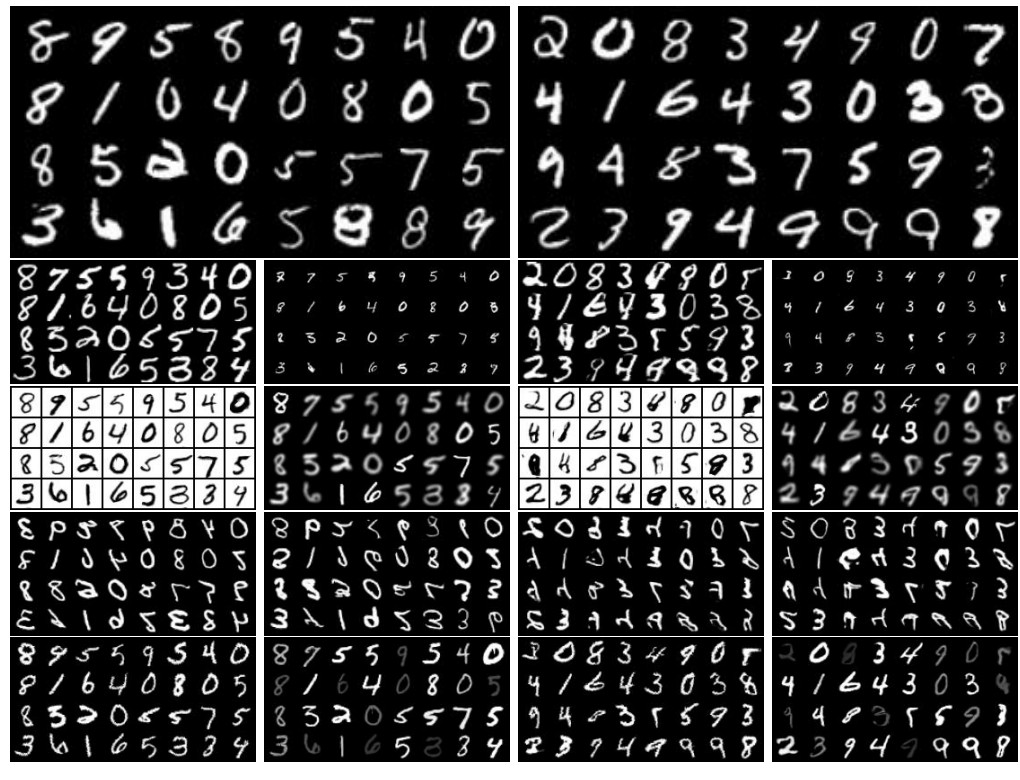

(a) Unconditional.                    (b) Conditional, 50% labels per client.

Figure 16: Transferring an image from MNIST's test set to all clients. Test image on top transferred image to each client at the bottom.

The effectiveness of this procedure can be seen in Figs. 15 and 16, where we transfer the content of some image to a specific client. In Fig. 15, we choose client 8 as the source and client 3 as the target to demonstrate that the reconstruction preserves both the content and style and that the transfer preserves the content. In Fig. 16, notice that, since the MNIST's test set is unseen during training, the unconditional GAN might map some seemingly easy digits to other similar-looking ones. Simply training on the test set should allow the unconditional GAN to mitigate this effect. Note that this effect is not seen in the conditional GAN, given that the clients see only 50% of the labels during training. It is worth mentioning that, as long as the content representation encoder $\phi_0^c$ is good enough for images from some unseen domains, like the test set of MNIST, then we can still transfer its content to a known client $i$ with a trained style vectorizer $G_i^s$.

### C.2.4 Experiments on VAEs

We consider applying our method on VAE ($\beta$-VAE to be specific), which is a well-known generative model that optimizes the evidence lower bound (ELBO), which is a general objective that many algorithms can be traced back to. Let $\mathbf{z} = (\mathbf{z}^c, \mathbf{z}^s)$, and with some abuse of notation, let $\theta^D, \theta^G$ be the parameters of the encoder and decoder, respectively. Then, the $\beta$-VAE objective can be written as:

$$f(\theta^D, \theta^G) = -\mathbb{E}_{\mathbf{z} \sim p_{\theta^D}(\mathbf{z}|\mathbf{x})} \log p_{\theta^G}(\mathbf{x}|\mathbf{z}) + \beta D_{\mathrm{KL}}(p_{\theta^D}(\mathbf{z}|\mathbf{x})||p_{\theta^G}(\mathbf{z})) \tag{16}$$

Perhaps the main difference here is that VAE's encoder outputs the latent variable's statistics, from which the latent can be sampled with the reparameterization trick.

The results can be seen in Fig. 8. One should be careful with applying the SSL regularizer with VAEs so that it does not overcome the training of the divergence term. Tuning VAEs is thus not very straightforward. Using lower values of $\beta$ and $\lambda \leq \beta$ seems to be desirable. We have not run extensive experiments on VAEs to obtain performance on par with our GAN experiments, but this setup should be a proof of concept that our

framework is possible to extend to other generative models and that its benefits are demonstrated, particularly for client 3 (color inversion). This is emphasized when taking into consideration the comparisons with Ditto in Fig. 7.

### C.3   CIFAR-10

Training a personalized GAN model on MNIST in the federated learning setting is in itself tricky, but we show that it is also possible on more realistic data, such as on the CIFAR-10 dataset. Similar results can be obtained on CIFAR-100, but we chose to show results only on CIFAR-10 for simplicity. The results for this experiment can be seen in Fig. 5 and Fig. 17.

For this experiment, we introduce 10 clients with different data augmentations and we train them using Adam with a local learning rate of 0.0003 and a global learning rate of 0.003. Precisely speaking, the data augmentations are

1. Zoom in: `CenterCrop(22)`.

2. Color inversion: `RandomInvert(p=1.0)`.

3. Blur: `GaussianBlur(5, sigma=(0.1, 2.0))`.

4. Grayscale: `Grayscale(3)`.

5. Vertical flip: `RandomVerticalFlip(p=1.0)`.

6. Brightness: `ColorJitter(brightness=0.8)`.

7. Perspective: `RandomPerspective(distortion_scale=0.4, p=1.0)`.

8. Solarize: `RandomSolarize(threshold=192.0)`.

9. Color Transform 1:

    (a) `Normalize(mean=[0.5], std=[0.5])`.
    (b) `transforms.Lambda(lambda x:  (x + 0.5 * torch.rand(3,1,1)) * torch.rand(3,1,1).clip(-1,1))`.

10. Color Transform 2:

    (a) `Normalize(mean=[0.5], std=[0.5])`.
    (b) `transforms.Lambda(lambda x:  (x + 0.5 * torch.rand(3,1,1)) * torch.rand(3,1,1))`.
    (c) `Normalize(mean=[0.5], std=[0.5])`.
    (d) `transforms.Lambda(lambda x:  x.clip(-1,1))`.

### C.4   CelebA

We test our model on CelebA, which is a more challenging dataset. The dataset contains pictures of celebrities, where each image is assigned a subset of attributes among 40 possible attributes. Thus, we create 40 clients, give a unique attribute to each, and then show them the subset with this attribute. Note that this is not a partition, as some data will be repeated many times, but we want to see if the client will learn a biased style towards the assigned attributes.

To make the simulation feasible, we create 10 workers with their models, each handling a unique subset of 4 clients, and let each worker sample their clients in a round-robin fashion. Thus, each worker will always see at least one of 4 attributes. We train our model for approximately 200 communication rounds, with 2 epochs per round, and train the discriminators 5 times as frequently as the generator (i.e. $T_D = 5$).

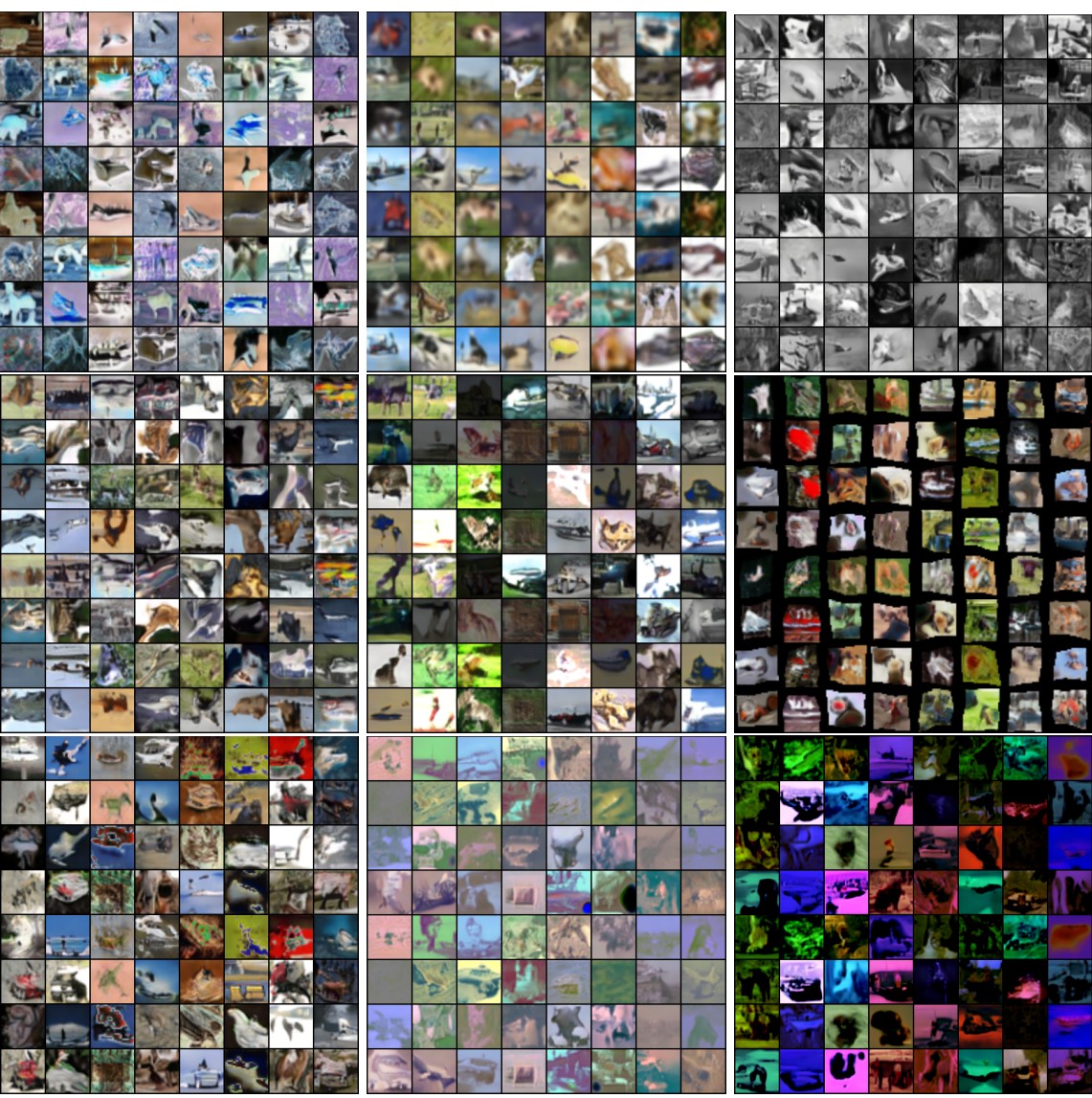

Figure 17: Samples from a non-conditional GAN on CIFAR-10 (skipping the first client corresponding to zoom-in augmentation). For each client, the first 4 rows share the same $\mathbf{z}^c$, while the last 4 rows share a different $\mathbf{z}^c$. Rows 1, 2, 5, and 6 share the same $\mathbf{z}^s$, while rows 3, 4, 7, and 8 share a different $\mathbf{z}^s$.

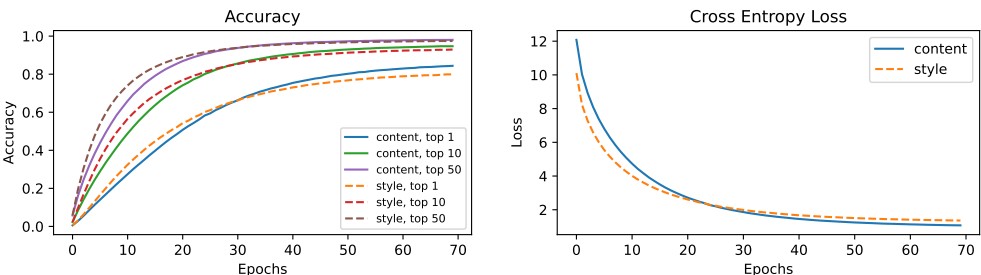

Figure 18: Accuracy and loss after training a linear classifier to predict the celebrity's identity. Note that there are 10,177 identities in total, which makes this task hard, so we also show top-10 and top-50 accuracies.

Given some content latent, we can see that the client-specific attributes become more obvious in some clients than others. For example, lipsticks, mustaches, blonde hair, and other attributes are clearer in some clients than in others. See Figs. 19 to 21 for samples from each worker.

For completeness and a better understanding of the generation quality, we list the attributes assigned to each worker:

1. Worker 1: ['Mustache', 'Mouth_Slightly_Open', 'Sideburns', 'Big_Lips'].

2. Worker 2: ['Attractive', 'Narrow_Eyes', 'Gray_Hair', 'Bald'].

3. Worker 3: ['Arched_Eyebrows', 'Bangs', 'Chubby', 'Eyeglasses'].

4. Worker 4: ['Male', 'Rosy_Cheeks', 'Wearing_Necktie', 'High_Cheekbones'].

5. Worker 5: ['Straight_Hair', 'Wearing_Earrings', 'Black_Hair', 'No_Beard'].

6. Worker 6: ['5_o_Clock_Shadow', 'Young', 'Wearing_Necklace', 'Wavy_Hair'].

7. Worker 7: ['Receding_Hairline', 'Bushy_Eyebrows', 'Goatee', 'Heavy_Makeup'].

8. Worker 8: ['Pointy_Nose', 'Blond_Hair', 'Double_Chin', 'Oval_Face'].

9. Worker 9: ['Big_Nose', 'Smiling', 'Blurry', 'Brown_Hair'].

10. Worker 10: ['Wearing_Lipstick', 'Pale_Skin', 'Bags_Under_Eyes', 'Wearing_Hat'].

**Celebrity identification from representation.** We also test the content and style features on a celebrity-identification task. The construction of the dataset is done so that we only test whether the style introduces a bias towards some attributes given some content, so we do not expect to see disentanglement between content and style. If we see one and only one attribute per client, then we might expect an attribute to be a style in the sense that it constitutes a domain, as in our MNIST example. In Fig. 18, we show the accuracy of a linear classifier on the content and style representations. Both achieve decent top-1 accuracies (84% and 79%, resp.), good 90% in top-10 accuracies (95% and 93%, resp.). Note that using top-10 and top-50 is approximately within the same proportion as using top-1 and top-5 for ImageNet, which has 1000 classes.

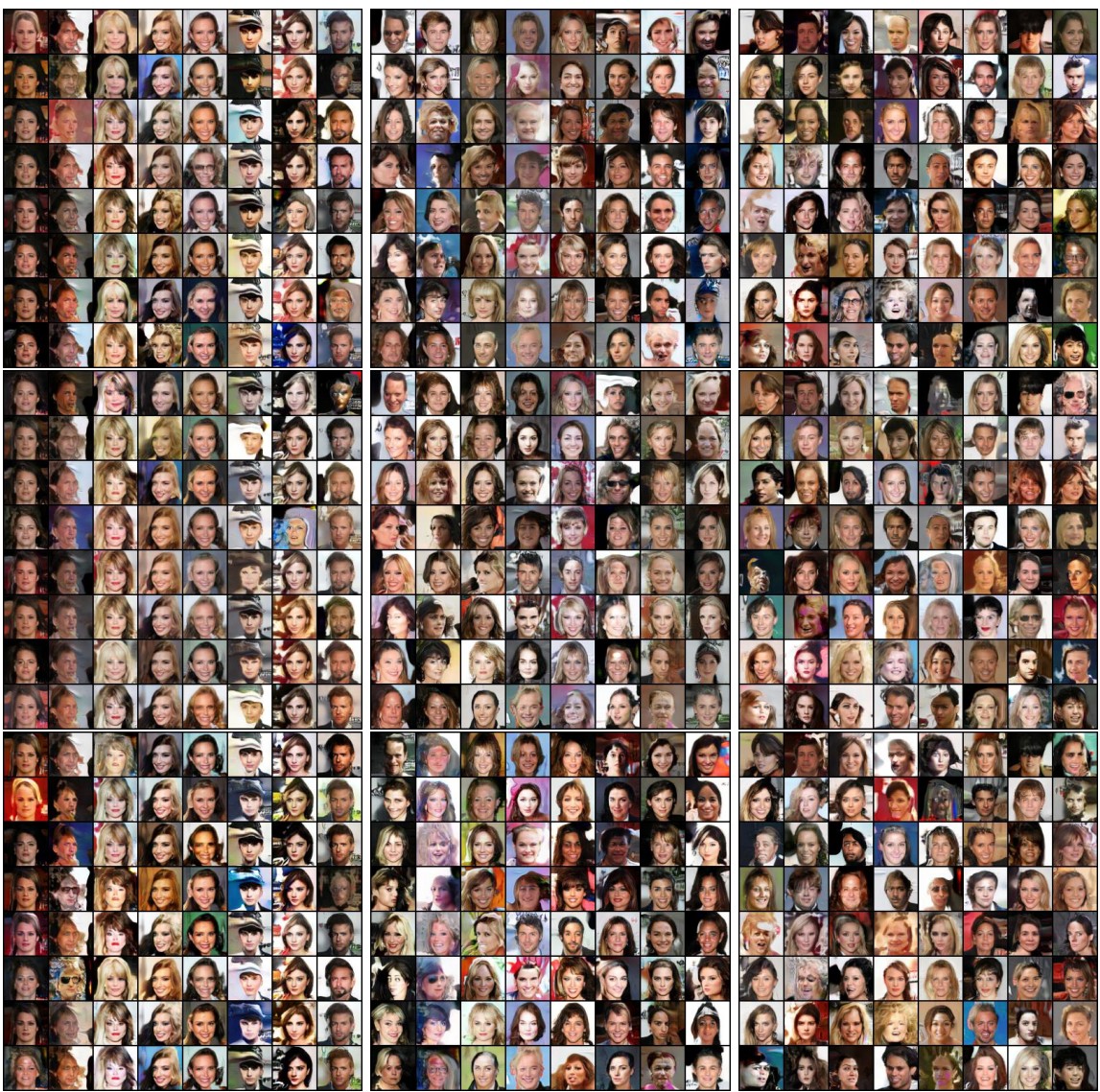

(a) Changing $\mathbf{z}^s$ per row, fixing $\mathbf{z}^c$. (b) Changing $\mathbf{z}^c$ per row, fixing $\mathbf{z}^s$. (c) Changing both $\mathbf{z}^c$ and $\mathbf{z}^s$.

Figure 19: Workers 1 at the top, 2 in the middle, and 3 at the bottom.

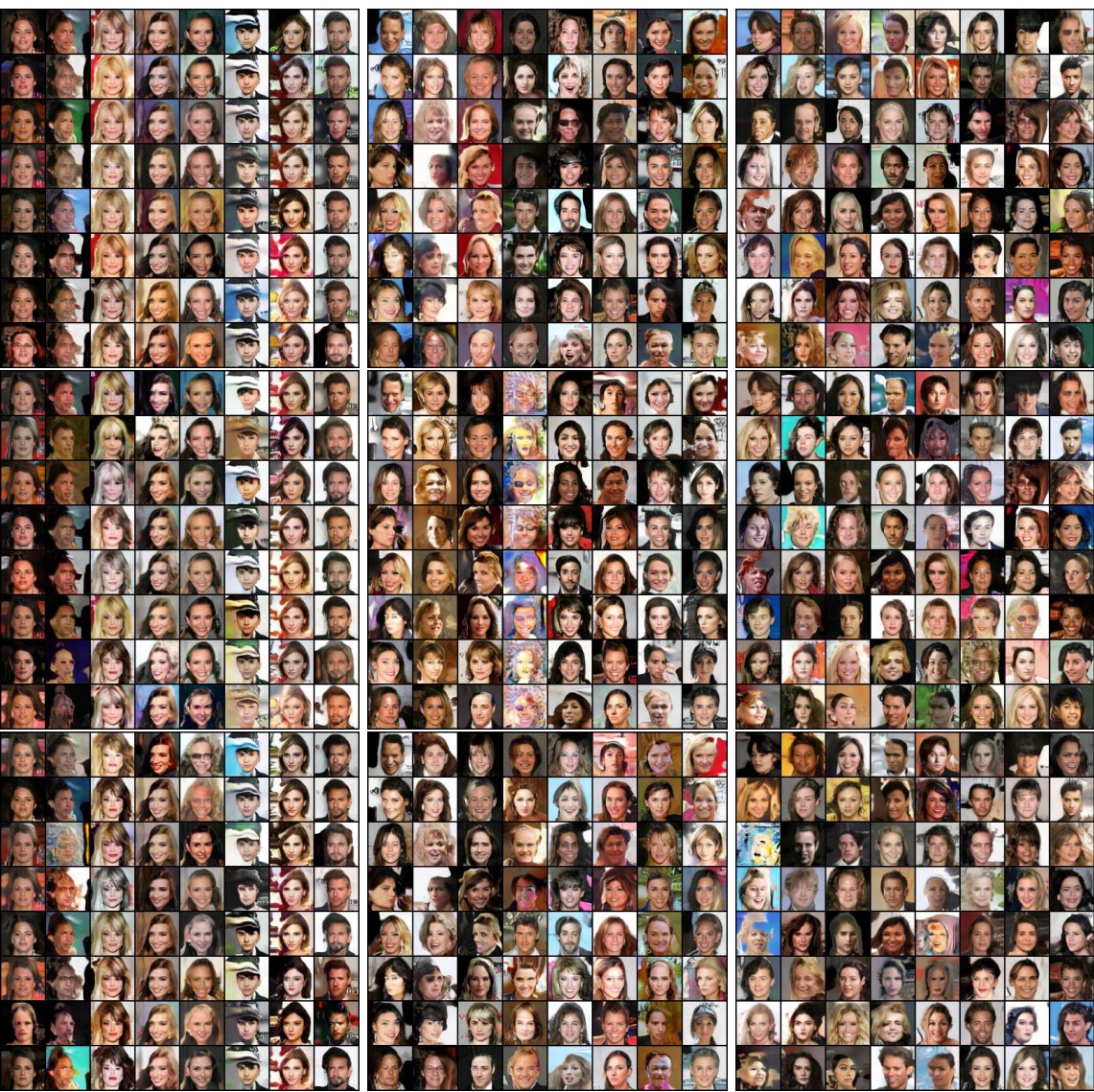

(a) Changing $\mathbf{z}^s$ per row, fixing $\mathbf{z}^c$. (b) Changing $\mathbf{z}^c$ per row, fixing $\mathbf{z}^s$. (c) Changing both $\mathbf{z}^c$ and $\mathbf{z}^s$.

Figure 20: Workers 4 at the top, 5 in the middle, and 6 at the bottom.

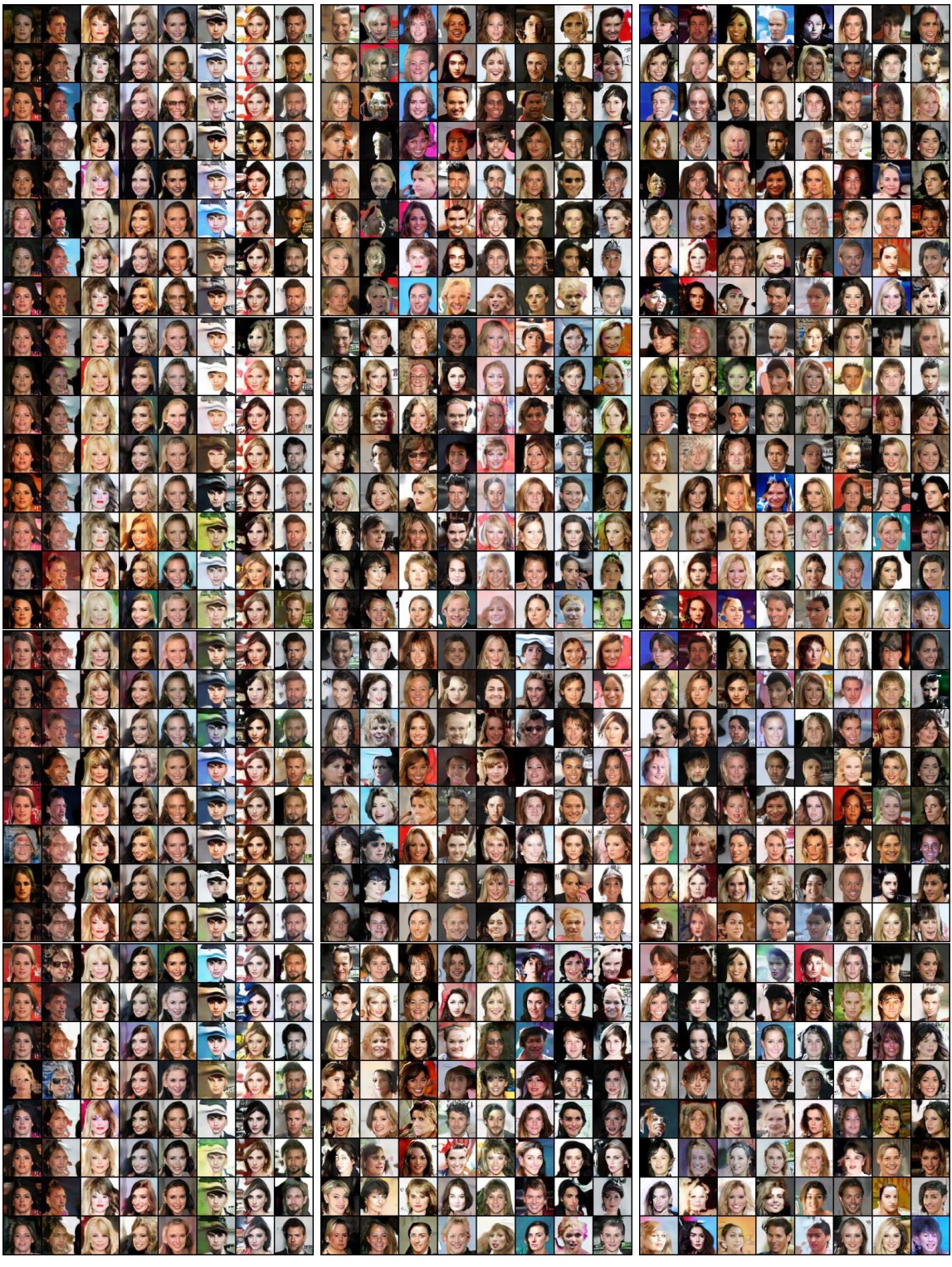

(a) Changing $\mathbf{z}^s$ per row, fixing $\mathbf{z}^c$. (b) Changing $\mathbf{z}^c$ per row, fixing $\mathbf{z}^s$. (c) Changing both $\mathbf{z}^c$ and $\mathbf{z}^s$.

Figure 21: Workers 7 at the top, 8 in the upper-middle, 9 in the lower-middle, and 10 at the bottom.

