# OpenReview forum: "PaDPaF: Partial Disentanglement with Partially-Federated GANs"
_TMLR — Accepted by TMLR_

### Review · Reviewer_gPAJ · 2023-12-08

**Summary Of Contributions:**

The paper introduces a novel framework for Partial Disentanglement with Partially-Federated Learning using Generative Adversarial Networks (GANs). In this approach, specific components of the model are federated, allowing the GAN's discriminator to partially disentangle representations into global and local factors. The primary focus is on global factors, crucial for inference. The disentanglement is reinforced by a "contrastive" regularization term inspired by the Barlow Twins methodology.

The proposed model achieves the ability to learn globally-consistent representations of data content, such as labels, leading to high-accuracy label classification. Notably, in supervised scenarios, the model demonstrates the capability to generalize to locally-unseen labels within individual clients concerning both classification and data generation.

Extensive experimentation supports the paper's claims, illustrating the seamless integration of techniques from federated learning, GANs, and representation learning. The results highlight the robustness of the model in real-world scenarios, showcasing its effectiveness in both classification tasks and data generation.

**Audience:**

Yes

**Claims And Evidence:**

Yes

**Requested Changes:**

- It would be great if the authors can explain whether the suggested method can be applied to generative models other than GANs. Some readers may want to apply this idea to more stable generative models, so it would be great to give some examples.
- The authors claim that "partial theoretical justifications for the proposed approach", but lack of some formal mathematical statement, i.e., Section 4 has some "discussions", but not proofs. I think it is better not to over-claim and tone down a bit regarding the theoretical contribution.
- Any baselines to compare empirically? Current experimental results are mostly showing the performance of the proposed method, but I want to know whether there are possibilities of comparing with existing methods that can be used for personalized generative models.

**Strengths And Weaknesses:**

- The idea makes sense and works well
- The theoretical understanding is not that sufficient

---

> ### Author Response · Authors · 2024-02-13
>
> We thank the reviewer for their comprehensive review and valuable suggestions and feedback.
>
> Regarding the experiments, other baselines, and generative models, we kindly ask the reviewer to read our response in the common reply above.
>
> > “The authors claim that "partial theoretical justifications for the proposed approach", but lack of some formal mathematical statement, i.e., Section 4 has some "discussions", but not proofs. I think it is better not to over-claim and tone down a bit regarding the theoretical contribution.”
>
> As we have only motivated our solution from a causal perspective, instead of having justified it, we have accordingly toned down the parts where we claim to “theoretically justify” or “theoretically validate” our proposed approach (specifically, in the abstract).

---

### Review · Reviewer_eMad · 2024-01-24

**Summary Of Contributions:**

The paper introduces a novel framework that aims to enhance federated learning with generative adversarial networks for personalized model generation. The framework combines global, client-agnostic models with local, client-specific models to achieve privacy and personalization.

**Audience:**

Yes

**Claims And Evidence:**

No

**Requested Changes:**

As in Weaknesses.

**Strengths And Weaknesses:**

Strengths：
+ It addresses the challenge of learning personalized generative models in a federated learning setting, which has been unexplored.
+ The paper achieves partial decoupling of content and style, allowing the model to distinguish between globally consistent features (content) and client-specific variations (style). This decoupling allows the model to share general knowledge across different clients while maintaining personalisation.
+ Experimental results show that the content features learnt through PaDPaF are able to support downstream tasks even in an unsupervised setting.



Weaknesses：
This paper could further be enhanced from the following perspectives:
- The experimental component lacks comparisons with other related work and there are no indicators to quantify the effect of the generating experiments
- The abstract section references the validation of the theory, yet this aspect is not adequately addressed or demonstrated within the main content of the text
- The paper mentions being able to address domain adaptation, but the treatment of data heterogeneity is not captured
- The paper discusses transformers in the introduction and also touches upon diffusion, which is deemed unnecessary for the context of this study.

---

> ### Author Response · Authors · 2024-02-13
>
> We thank the reviewer for their detailed review and critical feedback.
>
> Regarding the experimental component, we kindly point the reviewer to read the common response above.
>
> > “The abstract section references the validation of the theory, yet this aspect is not adequately addressed or demonstrated within the main content of the text”
>
> Indeed, we apologize for overclaiming. We have only motivated our solution from a causal perspective, instead of having “validated” it. We have edited this part accordingly (specifically, in the abstract).
>
> > “The paper mentions being able to address domain adaptation, but the treatment of data heterogeneity is not captured.”
>
> Our method can adapt to covariate shifts specifically, as we have assumed that p(y|x,i) is common among all clients i. It can also adapt to label shifts as can be seen from the experiments involving missing labels. We make no assumptions about the shift or heterogeneity itself other than the implicit assumption that the generator can capture this shift via its latent variables. This is why the adaptation is particularly good at stylistic differences, which is the crux of our method. The adaptation is clearly demonstrated on the MNIST and the new CIFAR-10 experiments, which is done unsupervised, but still demonstrate good downstream predictive performance (Figure. 5). Adaptation to unseen covariate shifts is also fast and shown to have good performance after as little as one round (see Appendix C.2.1).
>
> > “The paper discusses transformers in the introduction and also touches upon diffusion, which is deemed unnecessary for the context of this study.”
>
> We have addressed this point accordingly and removed this. Instead, we have added experiments on VAEs which we hope to be of interest to the reviewer (see Figure 8 and Appendix C.2.4).

---

### Review · Reviewer_kUTC · 2024-01-29

**Summary Of Contributions:**

The paper explores the potential of federated learning in personalized generative models. The authors propose a novel architecture that combines global client-agnostic and local client-specific generative models to achieve privacy and personalization in federated learning. The paper also highlights the benefits of using disentanglement in personalized generative models and provides experimental evaluation and theoretical justifications for the proposed approach.
Overall, the paper provides a comprehensive exploration of the potential of federated learning in personalized generative models. The proposed architecture is innovative and provides a promising approach to achieving privacy and personalization in federated learning. The experimental evaluation and theoretical justifications provided in the paper further strengthen the proposed approach's validity and potential for future research. However, the experiments are not sufficient. Basically, only two datasets are used. Meanwhile, more thorough literature review is also suggested.

**Audience:**

Yes

**Claims And Evidence:**

Yes

**Requested Changes:**

However, the experiments are not sufficient. Basically, only two datasets are used. Meanwhile, more thorough literature review is also suggested.

**Strengths And Weaknesses:**

Pros:
- The proposed architecture combines global client-agnostic and local client-specific generative models, which achieves privacy and personalization in federated learning.
- The use of disentanglement in personalized generative models provides benefits such as generating locally-unseen labels or attributes and removing private information.
- The experimental evaluation on various datasets shows the effectiveness of the proposed approach in disentangling content from style and achieving high accuracy on downstream tasks.
- The paper provides theoretical justifications for the proposed approach and suggests future research directions.

Cons:
- The experimental evaluation is limited to a few datasets (only two), and the generalizability of the proposed approach to other datasets or real-life applications is not fully explored.
- The paper does not provide a detailed comparison with other existing approaches in federated learning or personalized generative models. Basically, only GAN based works are discussed. Other possible generative models such as normalizing flow, VAE, Autoregressive, diffusion model works on Federated learning are not discussed and compared.
- Meanwhile, more thorough literature review is also suggested. Some SOTA federated learning papers are suggested to be included in the related work such as the following:
Personalized Federated Learning under Mixture of Distributions. ICML'23.

---

> ### Author Response · Authors · 2024-02-13
>
> We thank the reviewer for their mentioning the strengths of our work and providing us with their valuable feedback.
>
> Regarding the experimental evaluation and detailed comparison, we kindly ask the reviewer to read the common response above for our response.
>
> > “Meanwhile, more thorough literature review is also suggested. Some SOTA federated learning papers are suggested to be included in the related work such as the following: Personalized Federated Learning under Mixture of Distributions. ICML'23.”
>
> Thanks for sharing this interesting paper. We are familiar with this work, which is related to [1]. We did not discuss approaches based on clustering or mixture of distributions as we know by construction that each client has a distinct input distribution p(x) and a common labeling mechanism p(y|x). We are concerned with this setting, but we can see the application of clustering approaches when we have many clients that share the same heterogeneities (i.e. data augmentations). We understand the importance of this relevance, so we added a discussion of these works in the related work section.
>
> [1] Federated Multi-Task Learning under a Mixture of Distributions. Marfoq et al. 2021.

---

### Author Response · Authors · 2024-02-13
**Reply to All Reviewers**

We thank the reviewer for their insights and valuable suggestions. We appreciate that the reviewers have mentioned that this problem has been “unexplored”, and that our “novel” framework “makes sense and works well” and achieves “privacy and personalization”, with “experimental evaluations on various datasets” showing “the effectiveness of our proposed approach in disentangling content from style and achieving high accuracy on downstream tasks”.

Below, we show a common response to all reviewers, which mostly addresses problems in experimental evaluation, comparison to other baselines, and possibilities of extending the framework to other generative models. We show the relevant parts of each review towards which this response is aimed to answer.

**Reviewer kUTC**:
> “The experimental evaluation is limited to a few datasets (only two), and the generalizability of the proposed approach to other datasets or real-life applications is not fully explored.”

> “The paper does not provide a detailed comparison with other existing approaches [...] Other possible generative models [...] are not discussed and compared.”

**Reviewer eMad**:
> “The experimental component lacks comparisons with other related work and there are no indicators to quantify the effect of the generating experiments.”

**Reviewer gPAJ**:
> “Any baselines to compare empirically?”

> “It would be great if the authors can explain whether the suggested method can be applied to generative models other than GANs.”

**Response**:

According to the suggestions of the reviewers, we ran extra experiments on CIFAR-10, which is closer to a real-life application than MNIST. The experiments demonstrate similar improvements in personalized generation, and the benefits of the regularizer can be observed from the gap in prediction accuracy when using the content features vs. the style features. Experiment details can be found in the newly added Appendix C.3.
We also compare our method with Ditto [1] as a baseline, which is a well-known personalized FL method in the literature, and Ditto with FedProx [2]. We can visually verify in Figure 7 that our simple approach produces better images and show improvements in the FIDs per client (Table 1).

Furthermore, we have added experiments on VAEs. We chose VAE due to its simplicity as a generative model and the generality of its objective (see [3], for example). Our method can be generally applied to models that have an encoder and a decoder, where the encoder outputs two encodings or features (content and style) and the decoder takes a content latent and can be conditioned on a style latent. This construction is based on the causal model we assume (Figure 2). The regularizer is supposed to increase the “correspondence” between the content features and the content latent (i.e. make their correlation closer to identity w.r.t. some learnable metric) , which is desirable for downstream tasks (see assumption 1). The features of the discriminator in GAN models do not necessarily correspond to a latent variable, but we demonstrate that we can split the feature space such that one block corresponds to the content latent and the other to the style latent.

In case of VAEs, the encoding/feature space corresponds to the latent’s statistics (mean and variance), so it already corresponds to content and style accordingly and by construction. Hence, the regularizer can be redundant, but we can use it to ensure a better predictive performance of the features (keeping in mind that the regularization parameter should be smaller than ”beta”, or the weight of the KL divergence term). We added the results from our experiments in the revision of the paper (see Experiments section and Appendix C.2.4).

[1] Ditto: Fair and Robust Federated Learning Through Personalization. 2021. Li et al.

[2] Federated Optimization in Heterogeneous Networks. 2018. Li et al.

[3] Understanding Diffusion Objectives as the ELBO with Simple Data Augmentation. Kingma et al. 2023.

---

### Decision · Action_Editor_J5v8 · 2024-04-07

**Recommendation:** Accept as is

**Comment:**

The reviewers raised the following concerns about the original submission that:
1) it was not comparing sufficiently with the related work
2) the experimental evaluation was limited to only two datasets, and did not consider whether the approach could generalize to other generative models (e.g. VAEs)
3) some claims were not well supported (e.g. "theoretical justifications" in the abstract or the mention of transformers in the introduction as potential direction of future work)

All these concerns were addressed appropriately in the revision (see author's response):
1) they added comparison to related work (Ditto and FredProx)
2) they added one dataset (CIFAR) and also an experiment with VAEs, showing similar results
3) they removed the claims that were not well-supported.

Reviewer kUTC was happy with the changes and recommended a strong acceptance. The other two reviewers did not recommend acceptance, but justified their recommendation with the same arguments that they had put in the original review ignoring the important revision that the authors had uploaded (and not explaining why it was not sufficient). I have thus judged that the revision actually addressed their original concerns and that they should have updated their recommendation accordingly.

The authors should carefully proofread their paper for the camera ready version (perhaps with a spellchecker) as there are still some typos left.

**Audience:**

Yes.

**Claims And Evidence:**

Yes.